

# Toward Generalized Milankovitch Theory (GMT)

Andrey Ganopolski

Potsdam Institute for Climate Impact Research (PIK), Member of the Leibniz Association, P.O. Box 601203, D-14412, Potsdam, Germany

*Correspondence to*: Andrey Ganopolski (ganopolski@pik-potsdam.de)

**Abstract.** In recent decades, numerous paleoclimate records and results of model simulations provided strong support to the astronomical theory of Quaternary glacial cycles formulated in its modern form by Milutin Milankovitch. At the same time, new findings revealed that the classical Milankovitch theory is unable to explain a number of important facts, such as the change of the dominant periodicity of glacial cycles from 41 kyr to 100 kyr about one million years ago. This transition was

also accompanied by an increase in the amplitude and asymmetry of the glacial cycles. Here, based on the results of a hierarchy of models and data analysis, a framework of the extended (generalized) version of the Milankovitch theory is presented. To illustrate the main elements of this theory, a simple conceptual model of glacial cycles was developed using the results of an Earth system model CLIMBER-2. This conceptual model explicitly assumes the multistability of the climate-cryosphere system and the instability of the "supercritical" ice sheets. Using this model, it is shown that Quaternary glacial cycles can be

successfully reproduced as the strongly-nonlinear response of the Earth system to the orbital forcing, where 100 kyr cyclicity originates from the phase-locking of the precession and obliquity-forced glacial cycles to the corresponding eccentricity cycle. The eccentricity influences glacial cycles solely through its amplitude modulation of the precession component of orbital forcing, while the long time scale of the late Quaternary glacial cycles is determined by the time required for ice sheets to reach their critical size. The postulates used to construct this conceptual model were justified using analysis of relevant physical

and biogeochemical processes and feedbacks. In particular, the role of climate-ice sheet-carbon cycle feedback in shaping and globalization of glacial cycles is discussed. The reasons for the instability of the large northern ice sheets and the mechanisms of the Earth system escape from the "glacial trap" via a set of strongly nonlinear processes are presented. It is also shown that the transition from the 41 kyr to the 100 kyr world about one million years ago can be explained by a gradual increase in the critical size of ice sheets, which in turn is related to the gradual removal of terrestrial sediments from the northern continents.

The implications of this nonlinear paradigm for understanding Quaternary climate dynamics and the remaining knowledge gaps are finally discussed.

## 1 Introduction

Since the discovery of past glaciations in the mid-XIX century, the "ice-age problem" attracted significant attention and stimulated the first applications of physical science to understand climate dynamics. The idea that changes in Earth's orbital

parameters caused glacial ages was proposed soon after the discovery of ice ages (Adhemar, 1842) and has been further developed by a number of prominent scientists (see Berger, 1998, 2012 for the history of the astronomical theory of glacial



cycles). Milutin Milankovitch was one of them; he made an important contribution to the development of the astronomical theory of ice ages, published a hundred papers on this subject (e.g., Milankovitch, 1920), and presented his result in the most comprehensive form in his 650-page long "Canon of insolation and the ice-age problem" (Milankovitch, 1941). At present,

Milankovitch's version of the astronomical theory of ice ages is usually referred to as the Milankovitch theory. According to this theory, glacial cycles of Quaternary are forced by variations in boreal summer insolation, which in turn are caused by changes in three Earth astronomical ("orbital") parameters - obliquity, eccentricity, and precession.

During Milankovitch's life, this theory was just one of several hypotheses about the origin of past glaciations. Only during the 1970s spectral analysis of paleoclimate records confirmed the presence of periodicities predicted by the Milankovitch

theory, which was considered as the decisive proof of the astronomical theory. At the same time, analysis of paleoclimate data revealed some facts that the Milankovitch theory cannot explain. Among them are the dominance of 100 kyr cyclicity during the past million years, strong asymmetry of the late Quaternary (hereafter used as a synonym for the 100 kyr world) glacial cycles, and the change of the dominant cyclicity from 41 to 100 kyr around one million years ago. These findings stimulated numerous attempts to further develop the original Milankovitch theory or find alternative explanations for the mechanisms of

glacial cycles.

The problem of Quaternary glacial cycles has been approached from different perspectives using paleoclimate data analysis, development of simple (conceptual) models, and applying climate and ice sheet models of growing complexity. Despite significant progress in understanding climate dynamics and many publications devoted to modelling glacial cycles, the generally accepted comprehensive theory of glacial cycles did not emerge yet. However, one thing became increasingly

clear – the comprehensive theory of ice ages cannot be simple.

The present paper formulates a framework of the extended version of the Milankovitch theory of Quaternary glacial cycles, hereafter named Generalized Milankovitch Theory (GMT). The paper's main aim is to summarize the current progress in understanding and modelling Quaternary climate dynamics and facilitate further research in the field. The proposed theory is motivated and partly based on the results of the Earth system model of intermediate complexity CLIMBER-2 (Petoukhov et

al., 2000; Ganopolski et al., 2001), which so far is the only physically based model which can simulate glacial cycles during the entire Quaternary using orbital forcing as the only prescribed external forcing (Willeit et al., 2019). However, the theory presented here is not based on a single model and is thus not "model-dependent." On the contrary, this theory accommodates the results of a large amount of paleoclimate data analysis and numerous modelling studies. It is also important to note that this paper is not a review paper, and only the publications relevant to the theory presented in this paper are cited. The readers

can find a wealth of information about other works and alternative theories in review papers such as Berger (2012), Paillard, (2001, 2015), Berends et al. (2021).

The paper is organized as follows. Section 2 describes the classical Milankovitch theory and briefly reviews the works done to corroborate it. Section three is devoted to the conceptual models of glacial cycles. Section four presents a conceptual model of glacial cycles, which illustrates some essential aspects of the GMT. The fifth section presents a short discussion of

the main elements of GMT. In the conclusions, the main advances of GMT and the remaining challenges are discussed.



## 2.1 Original Milankovitch theory

Milankovitch theory is usually understood as a rather general concept that Quaternary glacial cycles were forced (or "paced") by changes in boreal summer insolation or, more specifically, that the Northern Hemisphere ice sheets were growing during periods of lower than average and shrinking during periods of higher than average boreal summer insolation. Thus,

irrespectively of how "summer insolation" is defined, it should contain contributions from obliquity and precession components, and the amplitude of the latter is modulated by the eccentricity (see Appendix A1). This is why, when all these frequencies were found in the late Quaternary paleoclimate records (Hays et al., 1976), this fact was widely considered as proof for the Milankovitch theory.

Less is known about the personal contribution of Milutin Milankovitch to the "Milankovitch theory" of ice ages. The most

widely known Milankovitch achievement is the meticulous calculations of insolation changes over the past million years. At the same time, the key premise of the Milankovitch theory, namely that glacial cycles are forced by boreal summer insolation changes, was not the original Milankovitch idea – in fact, Joseph John Murphy proposed it already in 1869 (Berger, 1988) and Milankovitch adopted this concept following advice from his friend and colleague Wladimir Köppen. The real contribution of Milutin Milankovitch to the development of glacial cycles theory was not in proposing a new hypothesis but rather in vigorous

testing of the existing hypothesis about the astronomic origin of glacial cycles. To this end, he calculated for the first time variations in insolation for different latitudes and seasons over the past 650,000 years accounting for all three Earth's orbital parameters: eccentricity, precession of the equinox, and obliquity. He then tested the astronomical theory by using a simple energy balance model, which also accounted for the effect of positive albedo feedback (Berger, 2021). Milankovitch also considered the influence of other potentially essential processes and the phase relationship between orbital forcing and

expected climate response. Finally, he attempted to validate theoretical predictions against available paleoclimate reconstructions and attributed individual ice ages known from geology (Fig. 1).

In the Milankovitch time, it was not known that the "glacial age" was the dominant mode of operation of the Earth system during the Quaternary. This is why Milankovitch considered ice ages as the episodes that occurred during periods of low summer insolation. Although Milankovitch did not explicitly formulate his conceptual model of glacial cycles, the text and

figures indicate that Milankovitch assumed that ice ages occurred during periods when caloric summer (half years) insolation was below a certain threshold value (Fig. 1). Using a simple energy-balance climate model, Milankovitch estimated that typical changes in summer insolation by 1000 canonical radiation units (1 canonical unit is approximately equal to 0.428 MJ) would cause 5°C summer cooling. Such cooling, in turn, is sufficient to lower the snow line by ca. 1 km and cause a widespread glaciation which is further amplified by the snow-






albedo feedback. Thus, Milankovitch, for the first time, demonstrated that variations in insolation caused by astronomical factors are capable of driving glacial cycles. According to the Milankovitch conceptual model, three glaciations occurred during the last 120 kyr, which at the time of Milankovitch were known as Würm I, II and III and now are usually notated as MIS 5d, 4 and 2. Although this was a significant step forward in explaining past glacial cycles, a comparison of the orbital forcing with the Earth system response shown in Fig.2 reveals problems with the classical Milankovitch theory. It turned out that the Milankovitch conceptual model is more applicable to the 41 kyr world of the early Quaternary rather than the 100 kyr world of the late Quaternary.

## 2.2 Testing Milankovitch theory with paleoclimate records

During the life of Milutin Milankovitch, no reliable paleoclimate records of glacial cycles existed to test his theory. Such records based on $\delta^{18}O$ in marine sediments became available only during 1960s (Emiliani, 1966), and already the first results of their analysis (e.g. Broecker and van Donk, 1970), provided strong support to the Milankovitch theory. Hays et al. (1976) demonstrated that the frequency spectra of planktic $\delta^{18}O$, considered the proxy for the global ice volume, contain both obliquity and precession frequencies, i.e., the frequencies of the Milankovitch insolation curves. However, the dominant periodicity of the recent glacial cycles was 100 kyr. This periodicity is absent in the spectrum of insolation but is close to one of the periodicities of eccentricity, which modulates the precession component of orbital forcing. In this landmark paper, the authors also demonstrated a close phase relationship between glacial cycles and eccentricity cycles and proposed that the 100 kyr cyclicity of the late Quaternary glacial cycles originates from a nonlinear climate response to eccentricity.

One of the most comprehensive attempts so far to test and extend Milankovitch theory beyond its original formulation was undertaken at the beginning of 1990th by a large group of scientists (also known as the SPECMAP group after the project acronym) in two papers - Imbrie et al. (1992) and Imbrie et al. (1993). In the first one, it was concluded that 23 kyr (precession) and 41 kyr (obliquity) cycles seen in different paleoclimate records can be explained as a linear response of the climate system to insolation changes in high latitudes of the Northern Hemisphere. In the second paper devoted to the origin of 100 kyr cyclicity in the late Quaternary, the authors discussed several potentially important processes and climate feedbacks but did not arrive at a definite conclusion about the nature of 100 kyr cycles. The SPECMAP group acknowledged that these cycles could originate from the existence in the climate system of internal self-sustained oscillations with a similar periodicity or, alternatively, 100 kyr cycles represent forced oscillations where the climate system "act a nonlinear amplifier which is particularly sensitive to eccentricity driven modulations in the 23,000-year sea level cycle" (Imbrie et al., 1993).

, The nature of glacial cycles and Milankovitch theory attracted significant attention in the following decades after SPECMAP publications, and numerous ideas of how to test and further develop Milankovitch theory have been proposed. First, the original Milankovitch theory has been turned from the theory of glacial ages into the theory of glacial terminations, and several studies analysed whether the timing of glacial terminations is consistent with the Milankovitch theory in its new termination version (e.g. Huybers and Wunsch, 2005; Raymo, 1997). Second, instead of caloric half-year summer insolation used by Milankovitch, the June, July, or summer solstice insolation at 65ºN became the standard metrics for the orbital forcing





(e.g., Berger 1978). Defined this way, the "orbital forcing" became dominated by precession, while in the original Milankovitch version, the contributions of obliquity and precession were comparable. Third, the Antarctic ice core analysis revealed significant variations in $CO_2$ concentration, which closely followed 100 kyr cycles of global ice volume (Petit et al., 1999). This led to the appreciation of the importance of $CO_2$ as the driver of glacial cycles and thus to the merging of the Milankovitch and Arrhenius theories, which before were considered to be alternative theories of glacial cycles (e.g., Ruddiman, 2003).

## 2.3 Testing Milankovitch theory with models

With the development of climate and ice sheet models, the efforts to test the validity of the Milankovitch theory also began on the modelling front. The impact of Earth orbital parameter changes was first investigated with atmospheric models (e.g., Kutzbach, 1981). It had been shown that surface air temperature and atmospheric dynamics (e.g., summer monsoons) are strongly affected by orbital parameter changes. Over the past twenty years, several "time slices" (mid-Holocene and the Eemian interglacial) have received special attention in the framework of paleoclimate intercomparison projects (e.g., Braconnot et al., 2007). Results of these studies confirmed the strong influence of orbital forcing on climate even though changes in Earth's orbital parameters have little effect on global annual mean insolation. It is noteworthy that a typical sensitivity of summer temperature over the Northern Hemisphere land to orbital forcing (about 1℃ per $10W/m^2$) found in climate model simulations is rather similar to the original Milankovitch estimate. Thus, the first premise of the Milankovitch theory of ice ages, namely, that changes in insolation caused by variations of the Earth's orbital parameters strongly affect climate, had been confirmed.

The next step in testing the Milankovitch theory was modelling climates favourable for glacial inception. The aim was to verify whether the cooling caused by the lowering of boreal summer insolation is sufficient to trigger large-scale Northern Hemisphere glaciation ("ice age"). Since the last glacial cycles began around 115 ka, the time when boreal summer insolation was about 10% lower than at present, most such experiments have been performed for this time. Simulations with different climate models show pronounced summer cooling over continents in the Northern Hemisphere. However, as far as the glacial inception (the build-up of ice sheets) was concerned, the results were rather ambiguous: some models simulated the appearance of perennial snow cover at least in several continental grid cells while others did not (e.g., Royer et al. 1983; Dong and Valdes 1995; Varvus et al., 2008). This ambiguity is not surprising since the magnitude of climate biases in model simulations are often comparable to models' responses to orbital forcing. In addition, a rather coarse spatial resolution of climate models used in these studies does not allow to resolve topographic details, which may be critical for the appearance of ice sheet nucleation centres (Marshall and Clarke, 1999). In any case, without coupling climate models to ice sheet models, it was impossible to determine whether the simulated climate response to orbital forcing was consistent with the observed ice volume growth rate.

Simulations of glacial cycles with simple ice sheet models began in the 1970s with 1-D models (e.g., Weertman, 1976; Oerlemans, 1980; Pollard, 1983). The latter study produces a rather impressive agreement between modelled and reconstructed ice volume. It also demonstrated the importance of the delayed isostatic rebound and what is now called marine ice sheet instability (MISI) for the realistic simulation of glacial cycles. Later, the complexity and spatial resolution of both climate and ice sheet modelling components increased, which allowed to simulate the temporal evolution of ice sheets during the past



glacial cycles and compare them with available paleoclimate reconstructions (e.g., Tarasov and Peltier, 1997; Berger et al.,
165 1999; Zweck and Huybrechts, 2003).

More recently, a new class of models, Earth system models of intermediate complexity (EMICs, Claussen et al., 2001),
began to be used for long transient simulations of the glacial cycles, first with prescribed GHGs concentration (e.g., Charbit et
al. 2007; Ganopolski et al., 2010; Heinemann, 2014; Choudhury et al., 2020) and finally with the fully interactive carbon cycle
(Ganopolski and Brovkin, 2017; Willeit et al., 2019). The latter study also succeeded in simulating the Mid-Pleistocene
170 Transition from the 41 kyr to 100 kyr world. One of the important results obtained with EMICs was the demonstration of the
existence of multiple equilibria states in the phase space of the orbital forcing and that the glacial inception represents a
bifurcation transition from one state to another (Calov and Ganopolski, 2005). Only recently transient simulations with ice
sheet models coupled to GCMs have become possible, but so far, such simulations are restricted to modelling only of glacial
inception (Gregory et al., 2012; Tabor and Poulsen, 2016; Lofverstrom et al., 2022).

## 3. Conceptual models of glacial cycles

Conceptual models of glacial cycles can be defined as a set of formal rules or mathematical operators that translate a certain
metric for orbital forcing (usually boreal summer insolation) into quantitative or qualitative information about the temporal
dynamics of glacial cycles. Despite the progress with the modelling of glacial cycles using the process-based model, conceptual
models remain a popular tool for studying the mechanisms of glacial cycles.

### 3.1 Qualitative conceptual model of glacial cycles

The term "qualitative" here is applied to the models that do not simulate the temporal evolution of climate characteristics but
instead provide only qualitative information about glacial cycles. The first model of this sort was proposed by Joseph Adhémar,
the author of the first astronomical theory of ice ages. He proposed that ice ages were caused by long winters and occurred
with the periodicity of the precession of the equinox, i.e., ca. 23,000 years (Adhémar, 1842). James Croll late changed long to
185 cold winters as the precondition for hemispheric glaciations (Croll, 1864). In turn, Milutin Milankovitch adopted Murphy's
view that glaciations occurred during "cold summers". In a quantitative form, Milankovitch's theory of glacial ages (see Fig.
1, which is similar to Fig. 48 from Milankovitch (1941)) can be formulated as follows: ice ages occur when summer insolation
drops below a certain threshold value.

Among examples of more recent qualitative conceptual models of glacial cycles is the idea that glaciations are triggered
by every fourth or fifth precession cycle after "unusually low" summer insolation maxima (Raymo, 1997; Ridgwell et al.,
1999). This conceptual model closely relates glacial cycles to the eccentricity cycles since "unusually low" summer insolation
maxima occur during periods of low eccentricity. An alternative conceptual model in which precession plays no role was
proposed by Huybers and Wunsch (2005). According to this model, glacial terminations of the late Quaternary were spaced in
time with equal probability by two or three obliquity cycles without any relation to other orbital parameters (see also Appendix
3). Admittedly, more recently, Huybers (2011) also accounted for the role of precession in pacing glacial cycles and proposed





a conceptual mathematical model where the timing of glacial termination is controlled by a metric of orbital forcing similar to Milankovitch's caloric half-year insolation.

Recently Tzedakis et al. (2017) proposed the rule which relates the timing of glacial terminations to the original Milankovitch metric for the orbital forcing, namely, half-year caloric summer insolation. Tzedakis et al. (2017) found that glacial terminations during the late Quaternary occurred when the "effective energy" (a function of caloric half-year insolation and elapsed time since the previous termination) exceeds a certain value. This conceptual model and its relationship to GMT are discussed in Appendix A3.

### 3.2 Mathematical conceptual models of glacial cycles

Mathematical conceptual models of glacial cycles are based on one or several equations that describe climate system response to orbital forcing. Usually, the state of the climate system is expressed through the global ice volume (e.g., Imbrie and Imbrie, 1980; Paillard, 1998), but some models also include equations for $CO_2$ concentration and temperature (e.g., Saltzman and Verbitsky, 1993; Paillard and Parrenin, 2004; Talento and Ganopolski, 2021) or some dynamical properties of the ice sheets (e.g. Verbitsky et al., 2018). Orbital forcing in such models is represented by a function of Earth's orbital parameters, usually by the maximum summer insolation at 65°N, caloric half-year insolation, or similar characteristics.

Calder (1974) and Imbrie and Imbrie (1980) proposed the first models of this type. Both models simulate global ice volume evolution in response to boreal summer insolation variations. While Calder's model simulates glacial cycles, which is dominated by precession and obliquity and has too little energy in the frequency spectra for the 100 kyr periodicity, the Imbrie and Imbrie model (see also Appendix A2) has too much energy in 400 kyr band, which is another eccentricity periodicity (see also Paillard, 2001). These problems are typical for many conceptual models. An important step in developing conceptual models of glacial cycles has been made by Paillard (1998) (P98 hereafter). This model is similar to the Imbrie and Imbrie model, but, in addition, it postulates the existence of different equilibrium states and different regimes of operation. This strong nonlinearity enables the model to achieve a very good agreement with paleoclimate reconstructions and, in particular, to eliminate the 400 kyr peak in the frequency spectra of ice volume. P98 model has played a critical role in the development of GMT and will be discussed in more detail in the forthcoming sections.

Admittedly, a number of other conceptual models simulate past glacial cycles with reasonable skill. In many of them, glacial cycles represent self-sustained oscillations with superimposed Milankovitch cycles (see Roe and Alen (1999) and Crucifix (2012) for the extensive discussion and comparison of different conceptual models). These studies also revealed that many models produce a similarly good agreement with the empirical data despite very different assumptions and formulations. Roe and Alen (1999) concluded: "We find there is no objective evidence in the record in favour of any particular model. The respective merits of the different theories must therefore be judged on physical grounds". To explain this apparent paradox, a "minimal" model of glacial cycles is described in the next section. The simplicity of the recipes needed to simulate "realistic" glacial cycles explains why different conceptual models produce similar results.





### 3.3 "Minimal" conceptual models of glacial cycles

One of the main challenges in modelling the late Quaternary glacial cycles is the dominant 100 kyr periodicity (a single sharp
peak in the frequency spectra at the period close to 100 kyr) and the absence of the energy maximum at 400 kyr, which is
another eccentricity period. While the linear transformation of orbital forcing would not show any significant energy at
eccentricity periods, nonlinear transformations (like Imbrie and Imbrie model, see Appendix A2) contain all eccentricity
periodicities. The simplest way to solve this problem is to design a model which possesses self-sustained oscillations with a
periodicity close to 100 kyr and where orbital forcing plays a secondary role, namely, the role of a pacemaker of long glacial
cycles. Such a model with appropriate parameters choice can simulate glacial cycles with the correct frequency spectrum and
timing of individual glacial cycles. Furthermore, such a model can be very simple and described by a single equation:

$$\frac{dv}{dt} = a_k - b\ f\ ,$$  (1)

where $v$ is the ice volume in arbitrary nondimensional units, time $t$ is in kiloyears, and orbital forcing $f = F\text{-}F_a$, where $F$ is the
maximum summer insolation at 65ºN and $F_a$ is the average value of $F$ over the last million years. The model has two "regimes":
k=1 corresponds to the period of ice growth and positive $a_1$, while k=2 corresponds to ice decay and therefore negative $a_2$. The
transitions from the regime k=1 to k=2 occur when $v$ =1 and from k=2 to k=1 when $v$ =0. In the absence of orbital forcing, the
solution is the sawtooth-like oscillations with the period $T= 1/a_1\text{-}1/a_2$. By proper choice of $a_1$ and $a_2$, this periodicity can be set
to 100 kyr. This is the "minimal" (MiM) model of glacial cycles which is similar but even simpler than the model described
in Leloup and Paillard (2022). This model has only three free parameters. For their optimal values given in Table 3, the model
can achieve a rather good agreement with the benthic $\delta^{18}O$ record (Lisiecki and Raymo, 2005) with the correlation coefficient
of 0.81 for the last 800 kyr (Fig. 3). Notably, the model solutions for the last 800 kyr strongly depend on the choice of model
parameters but does not depend on the initial condition if the simulations begin from $v$=0 not later than 1000 ka before present.
The model described by Eq. 1 is the simplest version of the relaxation oscillator, which is phase-locked to orbital forcing.
Many other conceptual models have more complex formulations but in terms of dynamics are essentially identical to the model
described by Eq. 1.

What can be learned from this simple modelling exercise? Obviously, it demonstrates that it is very easy to design a simple
model (mathematical transformation) which converts orbital forcing into the benthic $\delta^{18}O$ -like curve for the last 800 ka with
sufficiently good accuracy. These recipes are the following. First, the model has to possess two regimes: slow ice growth and
fast deglaciation regimes. Second, the transitions between the first and the second regimes should occur after some critical ice
volume is reached. Third, the time needed to reach this critical ice volume should be about 100 kyr. What remains to be
explained is

1) why the Earth system during Quaternary behaves like a relaxation oscillator;

2) what is the physical meaning of the "critical ice volume," and why reaching critical volume leads to regime change and
deglaciation;



3) why deglaciation is much shorter than the phase of predominant ice sheet growth;

    4) does the Earth system possess an internal time scale close to 100 kyr, or this time scale is directly related to the eccentricity.

    These and other questions will be discussed in Section 5. In the next section, a new conceptual model will be introduced and used to illustrate a number of essential aspects of GMT.

## 265  4. Representation of GMT in the form of a simple conceptual model

### 4.1 Modelling concept

As shown above, even a very simple conceptual model can simulate global ice volume evolution in good agreement with the reconstructed one. The reason why a new conceptual model is presented here is not that it has a better performance than other models of the same class but because it is based on the results of a process-based Earth system model CLIMBER-2. In turn,

it has been shown that CLIMBER-2 can successfully simulate Quaternary glacial cycles using orbital forcing as the only external forcing (Willeit et al., 2019). Thus, this new conceptual model is fundamentally different by design from other conceptual models, which are based on a subjective choice of their governing equations.

    The development of a new conceptual model has been done in two steps. First, based on the analysis of CLIMBER-2 model results (considered hereafter as Model 1), a simple but still physically based, reduced complexity model (Model 2) has

been designed. This model is described in Appendix A4. A 3-equations version of this model was used in Talento and Ganopolski (2021). Model 2 then has been additionally simplified, and in this way Model 3 was designed. This model is described below.

    The key results of CLIMBER-2, which have been obtained in Calov and Ganopolski (2005), Ganopolski and Calov (2011), and Willeit et al. (2019) were used to design Models 2 and 3 are the following:

1. The mass balance of ice sheets strongly correlates with the maximum summer insolation at 65°N.

    2. The existence of multistability of the climate-cryosphere system in the phase space of orbital forcing with the bifurcation transitions between different states.

    3. Typical (relaxation) time scale of the climate-cryosphere is about 20-40 kyr.

    4. Robustness of simulated glacial cycles regarding the choice of initial conditions and model parameters.

5. Phase locking of the late Quaternary glacial cycles to the 100 kyr eccentricity cycle.

    6. Strong asymmetry between ice sheet growth phase and glacial terminations.

    Model 3 does not explicitly account for the role of $CO_2$ in glacial cycles, which was analysed in our previous studies (e.g. Ganopolski and Calov, 2011; Ganopolski and Brovkin, 2017). For the sake of simplicity, it is assumed here that the role of $CO_2$ is implicitly represented by ice volume since these two characteristics are highly correlated. The role of $CO_2$ in shaping

glacial cycles will be discussed in the next section.



## 4.2 Model 3 formulation

The new conceptual model of glacial cycles (Model 3) is based on the existence of multiple equilibrium states found in Calov
and Ganopolski (2005) and reproduced by Model 2 (see Appendix A4). The first (interglacial) state is characterized by a warm
climate and the absence of continental ice sheets in the Northern Hemisphere. The second one, with the massive ice sheets
covering significant fractions of North America and Eurasia, is the glacial state. In the framework of this concept, the evolution
of the Earth system under the influence of orbital forcing is described as the relaxation towards the corresponding equilibrium
state. Model 3 contains only one variable – global ice volume $v$ (in nondimensional unit) described by the following equation:

$$\frac{dv}{dt} = \begin{cases} \dfrac{V_e - v}{t_1}, & k = 1 \\[2ex] \dfrac{v_c}{t_2}, & k = 2 \end{cases}$$
(2)

with the additional constrain $v \geq 0$. Equation (2) describes two different regimes of ice sheet evolution. The first one (k=1),
similar to the P98 model, is the glaciation regime when the system relaxes toward the equilibrium glacial state $V_e$ with the time
scale $t_1$. The second regime (k=2) is the deglaciation regime when ice volume linearly declines toward zero with the
characteristic time scale $T_2$, with $v_c$ being the model parameter that defines the critical ice volume (see below). The equilibrium
state $V_e$ towards which the system is attracted is a function of orbital forcing and, for the bi-stable regime, also depends on ice
volume (see Fig 4a):

$$V_e = \begin{cases} V_g, & if\ \ f < f_1,\ \ or\ \ \ f_1 < f < f_2,\ \ and\ \ v > V_u \\[2ex] V_i, & if\ \ f > f_2,\ \ or\ \ \ f_1 < f < f_2,\ \ and\ \ v < V_u \end{cases}$$

The glacial equilibrium state is defined as

$$V_g = 1 + \sqrt{\frac{f_2 - f}{f_2 - f_1}}\ ,$$

the unstable equilibrium which separates the glacial and interglacial attraction domains is defined as

$$V_u = 1 - \sqrt{\frac{f_2 - f}{f_2 - f_1}}\ ,$$

and the interglacial equilibrium $V_i = 0$. Here orbital forcing $f$ (in W/m$^2$), similar to the minimal conceptual model MiM, is
defined as $f = f - f_a$, where $f$ is the maximum summer insolation at 65°N and $f_a$ is its averaged value over the last million years,
$f_1$ and $f_2$ are model parameters. Note that orbital forcing only enters Eq. (2) in a parametric form.





The transition from glacial (k=1) to deglaciations regime (k=2) occurs if three conditions are met: $v > v_c$, $\dfrac{df}{dt} > 0$ and $f >$ 0. The transition from deglaciation to glacial regime occurs if orbital forcing $f$ drops below the glaciation threshold $f_1$. The interglacial state formally belongs to the deglaciation regime.

The model has five parameters ($t_1$, $t_2$, $f_1$, $f_2$, $v_c$), all of which, in principle, can be used to maximize the agreement between simulated and reconstructed ice volume. However, all these parameters have clear physical meaning and can be independently estimated using the results of CLIMBER-2 and paleoclimate data. The value of $f_1$ (insolation threshold for glacial inception) is rather tightly constrained by the current insolation minimum ($f$=-15 W/m²) when glaciation did not occur and MIS19 insolation minimum ($f$=-20 W/m²) when it did occur (Ganopolski et al., 2016). According to Calov and Ganopolski (2005),

the relaxation time scale $t_1$ is about 30 kyr, and $f_2$ is positive. It is noteworthy that model results depend on a combination of $t_1$ and $f_2$, and essentially identical solutions can be obtained for different combinations of these two parameters. Deglaciation time scale $t_2$ derived from model and paleoclimate records is about 10 kyr. The last model parameter, the critical ice volume $v_c$ controls the dominant periodicity and degree of asymmetry of glacial cycles. As a result, only $f_2$ and $v_c$ were used as tunable parameters and their values (Table 1) have been chosen to maximise the correlation between simulated ice volume and benthic

$\delta^{18}O$ record (LR04) during the past 800 kyr. The $\delta^{18}O$ has been used here as the ice volume proxy even though several reconstructions of sea level for the last 800 kyr exist (i.e. Spatt and Lisiecki, 2016). This has been done to enable the comparison of model results with the paleoclimate data for the entire Quaternary. Such choice is justified by a strong similarity between the LR04 stack and late Quaternary sea level reconstructions. Model 3 is similar to the P98 model by formulation and its results, but it does not contain time scales close to 100 kyr.

**4.3 Simulation of the Late Quaternary glacial cycles with the Model 3**

Results of model simulations depicted in Fig. 5 show a good agreement with the LR04 stack for the last 800 kyr. The correlation between model and data is 0.85 for the entire time interval. The agreement is better for the later part (the correlation is 0.9 when only considering the last 400 kyr) than for the earlier part, which can be partly explained by the fact that the model and LR04 are in antiphase around 700 and 490 kyr. Since LR04 has been tuned to the Imbrie and Imbrie model, which has a very

strong precession component, the cause for such a large data-model mismatch is unclear. The frequency spectrum of the model results in good agreement with paleodata has a dominant sharp 100 kyr maximum, with little energy in 400 kyr band and a relatively weak precession component compared to the spectrum of orbital forcing (Fig. 5c). Simulated glacial cycles are rather insensitive to the initial conditions since they converge rapidly to the common solution (Fig. 6c). This agrees with CLIMBER-2 simulations (Willeit et al., 2019) where we concluded that "simulated glacial cycles only weakly depend on initial conditions

and therefore represent a quasi-deterministic response of the Earth system to orbital forcing". Modelling results are also robust regarding the choice of model parameters. For example, when keeping other model parameters constant, as in Table 1, essentially the same results are obtained for the range of $v_c$ between 1.32 and 1.49 (Fig. 6b). It is noteworthy that, although Model 3 is based on CLIMBER-2 and aimed to mimic it, Model 3 actually outperforms the CLIMBER-2 model in terms of





the agreement with empirical data for the last 800 kyr. In particular, CLIMBER-2 has a problem with simulating the correct
timing of Termination V prior to MIS 11 (see Willeit et al., 2019), while Model 3 does not have such a problem.

The existence of two regimes of operation has been postulated in Model 3 similarly to P98, and it is absolutely critical for simulating realistic glacial cycles. To demonstrate this, an additional experiment has been performed with Model 3 but without the termination regime. This experiment started from $v$=0 at 125 ka (Eemian interglacial) and was run to present day. The trajectory of ice volume evolution in phase space of orbital forcing for this experiment is compared with the standard Model 3
in Fig.4b. The two model versions are identical before 20 ka when the termination regime is activated in the standard model version. If the termination regime is disabled, the model trajectory in the phase space represents a loop similar to previous precession cycles (green line) and does not result in deglaciation. This result does not dependent of the choice of model parameters, and there are no combinations of model parameters that enable the simulation of the realistic glacial cycles without the termination regime.

**4.4 Generic orbital forcing and the origin of 100 kyr cyclicity in Model 3**

In Ganopolski and Calov (2011) we argued that "100 kyr peak in the power spectrum of ice volume results from the long glacial cycles being synchronized with the Earth's orbital eccentricity". To understate how this synchronization occurs, Model 3 was forced by a generic orbital forcing instead of the real summer insolation. This generic forcing consists of a periodic precession-like harmonic component with a single periodicity of 23 kyr, which amplitude is modulated by schematic
eccentricity-like cycles with the periodicity of 100 kyr:

$$f = A\left(1 + \varepsilon \sin \frac{2\pi t}{100}\right)\cos\left(\frac{2\pi t}{23}\right), \tag{3}$$

where $A$ is the magnitude of forcing in W/m$^2$ and $\varepsilon$ is the nondimensional magnitude of amplitude modulation.

The first experiment is performed with the simplest periodic orbital forcing with A=25 W/m$^2$ and $\varepsilon$=0. Fig 7 shows model results for three values of the critical ice volume: $v_c$=1.2, 1.33 and 1.47. The model simulates periodic glacial cycles with the
amplitudes and periods increasing with $v_c$. Naturally, these periods are multiples of 23 kyr. For $v_c$ =1.2 this period is equal to 3x23=69 kyr, for $v_c$ =1.47 the period is 6x23=140 kyr, and only for $v_c$ =1.33 the period 4x23=92kyr is relatively close to 100 kyr (Fig 7c). Note that all these periods are much longer than the relaxation time scale of the model equal to 30 kyr.

Applying amplitude modulation ($\varepsilon$=0.5) with a periodicity of 100 kyr to this generic forcing, leads to quasiperiodic cycles with a sharp peak at 100 kyr in the frequency spectrum and similar timings of glacial terminations for all three values of critical
ice volume (Fig. 7b, d, f). This is because, irrespectively of the value of $v_c$, all simulated cycles are phase locked to the amplitude modulation cycle. This result is very robust with respect to the $v_c$ value and the amplitude and period of precession-like cycles, as well as to the amplitude and periodicity of the amplitude modulation cycle. Similarly, setting the periodicity of precession-like cycles in the range from 10 to 30 kyr, has a minimal impact on the results (not shown).





The mechanism of the phase locking of long glacial cycles to the 100 kyr amplitude modulation cycle in the case of the

generic orbital forcing is rather straightforward. After ice volume $v$ exceeds a value of about $0.2v_c$, the system stays longer in the attraction domain of the glacial state (see Fig. 4). Moreover, the smaller the amplitude of orbital forcing, the longer it stays in the attraction domain of the glacial state when ice is growing. Thus, the likelihood of reaching the critical ice volume $v_c$ is higher during periods of weak orbital forcing, i.e. during periods of low eccentricity. It is also noteworthy that the amplitude of simulated glacial cycles and the height of 100 kyr peaks in the frequency spectrum of ice volume (Fig. 7) is essentially

independent of the magnitude of amplitude modulation in a wide range of parameter $\varepsilon$ values.

## 4.5 Is the spectrum of ice volume variability consistent with the phase locking of glacial cycles to eccentricity?

While the formal relationship between eccentricity and late Quaternary glacial cycles has been established already in Hays et al. (1976), some studies (e.g., Muller and MacDonald, 1997; Maslin and Brierley, 2015) argued against the direct link between 100 kyr cyclicity of glacial cycles and eccentricity. It is known that the direct effect of eccentricity on global insolation is

negligibly small (e.g. Paillard, 2001), and this is why the eccentricity can affect climate only through the amplitude modulation of the precession cycle. In principle, the response of any nonlinear system to the forcing, which consists of a quasi-periodic carrier with amplitude modulated by another quasi-periodic signal, should contain periodicities of both carrier and amplitude modulation signal. However, eccentricity also has a strong 400 kyr periodicity which is practically absent in the late Quaternary ice volume reconstructions. In addition, the peak in the eccentricity spectrum near 100 kyr is split into two peaks at 95 and 124

ka, while the spectrum of the reconstructed ice volume contains very little energy at 124 kyr.

Results of the Imbrie and Imbrie model apparently support this critique since the spectrum of their model reveals all problems mentioned above: the low-frequency part of its spectrum is essentially identical to the spectrum of eccentricity (Fig. 8d) and very different from $\delta^{18}O$ spectrum for the last 800 kyr (Fig. 3). However, it is possible to construct a mathematical transformation that converts orbital forcing into realistic ice volume evolution, and Model 3 is an example of such

transformation (Fig 5). To enhance the resolution of frequency spectrum, the model has been run through the past 2.8 Ma with the fixed model parameters. Unsurprisingly, the model with parameters tuned to the 100 kyr world cannot reproduce the 41 kyr world, and simulated glacial cycles are realistic only for the last million years. Unlike the Imbrie and Imbrie model, the spectrum of Model 3 does not contain a 400 kyr peak and has a single sharp peak at 95 kyr (Fig. 8d). Such behaviour is robust for a wide range of model parameters, for example, for $v_c$ in the range between 1.2 and 1.5. The absence of 400 kyr periodicity

in the frequency spectra of Model 3 can be explained by the fact that each glacial termination "erases" the system memory, and the amplitude of the next glacial cycle does not depend on the previous one. The main reason why the spectrum of Model 3 is so different from the spectrum of eccentricity is because the eccentricity is not the forcing of glacial cycles, but rather a pacemaker that sets the dominant periodicity and, to some degree, the timing of glacial terminations.

Although the spectrum of simulated glacial cycles is dominated by a sharp peak at around 100 kyr, this does not imply that

all glacial cycles have a duration close to 100 kyr. In fact, as Fig. 8e shows, the duration of individual glacial cycles tends to cluster in the intervals of 80-90 and 110-120 kyr, which are both close to the duration of 2 or 3 obliquity cycles and 4 or 5



precession cycles. On the other hand, most durations are between 80 and 120 kyr, and very few are outside, which would not be the case if glacial cycles will simply last 2 or 3 obliquity cycles. This feature is also seen in the distribution of durations of the last eight real glacial cycles. Also, as in reality, very few simulated glacial cycles have durations between 90 and 110 kyr.

This explains the apparent paradox of why none of the glacial cycles of the 100 kyr world has a duration close to 100 kyr.

Another apparent paradox related to the role of eccentricity in glacial cycles is related to the fact that the energy in 100 kyr band of ice volume spectrum was growing over the past million years while the energy in 100 kyr band of eccentricity was decreasing during the same time (Lisiecki 2010; see also Fig. 9). Partly this discrepancy can be explained by the fact that the energy increase in the 100 kyr band of ice volume spectra during the Mid-Pleistocene Transition (MPT, ca. 1.2-0.8 Ma) was

related to changes in the boundary conditions (Willeit et al., 2019). However, even when keeping model parameters constant (Fig. 9), the energy in the 100 kyr band of ice volume spectrum changes in antiphase with the energy in the eccentricity spectrum. This can again be explained by the fact that eccentricity is not the direct forcing of glacial cycles, and the amplitude of glacial cycles is not directly related to the amplitude of the 100 kyr component of eccentricity, which is also clearly seen in the experiments with the generic orbital forcing (Fig. 7).

**4.6 Simulation of the Mid-Pleistocene transition**

When applied to the entire Quaternary, Model 3 with constant parameters tuned to the last million years fails to reproduce the glacial cycles of the early Quaternary (Fig. 9). This is consistent with Willet et al. (2019), where it was shown that orbital forcing alone could not cause the regime change million years ago. To reproduce the MPT in P98, the critical ice volume was made time-dependent with a smaller value at the beginning of Quaternary and a larger one toward the present. A similar

approach works for Model 3 too. The critical volume $v_c$ is a nonlinear function of time:

$$v_c = 0.5(v_{c1} + v_{c2}) + 0.5(v_{c2} - v_{c1})\tanh\left((t - t_t)/\tau_t\right), \tag{4}$$

where $t$ is time in kiloyears before present (negative), $t_t = -1050$ kyr is the centre of MPT, transition time $\tau_t = 250$ kyr, and the initial and final critical ice volume values $v_{c1}=0.65$, and $v_{c2}=1.38$. Notably, this temporal evolution of $v_c$ closely resembles the temporal evolution of the regolith-free area prescribed in Willeit et al. (2019). Such similarity is in line with the idea that the

temporal evolution of the critical size of ice sheets is controlled by sediments removal from northern continents by glacial erosion processes.

With such time-dependent $v_c$, the model can reproduce a significant increase in the amplitude of glacial cycles and the transition from obliquity dominated 41 kyr to 100 kyr dominated cycles around one million years ago. The agreement between model results and LR04 before 2Ma is rather poor but it improves significantly after 2 Ma (Fig. 10b), and the correlation

between modelled ice volume and $\delta^{18}$O over the last 2.7 Ma is 0.75. The wavelet spectrum of ice volume also shows a good agreement with the spectrum of LR04 stack (Fig. 10c, d). Prior to the MPT, the spectrum of modelled ice volume is dominated by obliquity even though the orbital forcing is dominated by precession. Still precession component is significantly stronger in model output compared to LR04. This issue will be discussed in the next section and Appendix A6. Similar to LR04 and

CLIMBER-2 results (Willeit et al., 2019), the maximum energy in the wavelet spectrum in Model 3 first moves from 40 to 80

kyr (two obliquity cycles) at ca. 1.2Ma, and only after 0.8 Ma does the 100 kyr periodicity become the dominant one. Also, in agreement with paleodata, the period of energy maximum during the last 0.8 Ma does not remain constant and shows a clear tendency to oscillate around 100 kyr with a 400 kyr periodicity, which is another eccentricity period.

## 4.7 Glacial cycles in Model 3

Model 3 represents a rather simple nonlinear transformation of the traditional orbital forcing into glacial cycles. An important

feature of this model is that it is based on simulation results using the Earth system model of intermediate complexity CLIMBER-2. The key element of the model is the existence of two fundamentally different regimes: glacial with the relaxation towards one of two equilibrium states and the deglaciation regimes. The model does not possess self-sustained oscillations and has no intrinsic time scale close to 100 kyr. In fact, this period originates solely from the phase locking of long glacial cycles to the amplitude modulation of precession components of insolation by eccentricity. The model not only accurately reproduces

the temporal evolution of the glacial cycles, including the timing of terminations, but also the frequency and wavelet spectra of the late Quaternary. The model helps to understand how a dominant 100 kyr periodicity originates from the combinations of different eccentricity, obliquity and precession periods. In the next section, I will discuss how such mathematical transformation can be explained from the "physical" point of view.

## 5. Elements of GMT

This section presents the key elements of the GMT; namely, it discusses physical and biogeochemical processes which play a crucial role in Quaternary glacial cycles. It also explains the postulates which make Model 3 so successful. Unlike the simplicity of the conceptual model, the physical basis of the GMT is very complex, and some important processes and mechanisms are still not fully understood. This is why some aspects of the GMT are only preliminary.

## 5.1 Orbital forcing

When developing his theory, Milankovitch adopted the already proposed idea that changes in boreal summer insolation are critical for understanding ice ages. As the metric for summer insolation, Milankovitch used the integral of insolation during a warm (caloric) half-year. After the revival of interest in the Milankovitch theory in the 1970s, it became more common to use the maximum (or summer solstice), June or July insolation at 65°N to analyze the relationship between orbital forcing and climate system response. When discussing which single metric best represents orbital forcing, it is important to realize that

climate models calculate insolation at each time step and for each grid cell, and therefore climate modellers do not need to decide what definition of "summer insolation" is correct. However, for analysis of paleoclimate records and construction of simple (conceptual) models of glacial cycles, it is required to convert 3D (latitude – day of the year – the year before/after present) insolation into a single metric. This paper uses maximum summer insolation at 65°N for this purpose, and this choice is discussed in Appendix A1.



Irrespective of the specific choice of the metric for "orbital forcing", any "summer insolation" curve represents a sum of precession and obliquity components. The precession component is a quasiperiodic (T≈23,000 years) sine-like function with an amplitude proportional to eccentricity. The obliquity component is a periodic (T≈41,000 years) function with the amplitude gradually changing in time. The relative contributions of precession and obliquity components depend on the choice of latitude and definition of "summer," but, as shown in Appendix A1, it is reasonable to assume that obliquity and precession components

are of comparable importance. The only notable exception is the so-called "summer energy" proposed in Huybers (2006), containing very little precession. However, as shown in Appendix A1, "integrated summer energy" is not applicable to the problem of glacial cycles.

## 5.2 Climate feedbacks

Experiments with climate models which began in the 1970s confirmed the central premise of the Milankovitch theory, namely,

that variations of Earth's orbital parameters cause significant (several degrees and more) regional summer temperature changes over the continents. Such temperature changes would result in large vertical displacements of the equilibrium snow line and can, in principle, cause widespread glaciation of some regions. However, it has been found that the direct impact of the orbital forcing on climate in terms of global mean temperature is very small (less than 1°C). Thus, to understand how orbital forcing during Quaternary cased global-scale large-amplitude climate oscillations, it is necessary to invoke several climate-related

feedbacks.

    The first important positive feedback is the albedo feedback, which is already accounted for in Milankovitch calculations (Milankovitch, 1941). Results of modelling studies demonstrated that, even under a constant but sufficiently low $CO_2$ concentration (lower than the typical interglacial level of 280 ppm), orbital forcing alone could drive glacial cycles with realistic amplitude and periodicities (Berger and Loutre, 2010; Ganopolski and Calov, 2011; Abe-Ouchi et al. 2013). In turn,

large continental ice sheets strongly affect global temperature. According to model simulations, ice sheets and associated sea level drop explain about half of the global cooling at LGM (Hargreaves et al., 2007). However, this effect caused by albedo and elevation changes over large continental ice sheets is rather local and diminishes rapidly with the distance from the margins of ice sheets. Moreover, in some regions, due to the modifications of atmospheric circulation, the effect of ice sheets on temperature can be even opposite, i.e., the growth of ice sheets can cause regional warming (Liakka and Lofverstrom, 2018).

The main "globalizer" of glacial cycles is $CO_2$, while methane and $N_2O$ contribute together to the radiative forcing of glacial cycles about 1/3 of $CO_2$. Interestingly, the role of $CO_2$ in glacial cycles was proposed by Svante Arrhenius (Arrhenius, 1896) well before Milankovitch published his first paper. Although Milankovitch was not enthusiastic about Arrhenius's theory, it is generally accepted now that both the Milankovitch astronomical theory and Arrhenius' $CO_2$ theory represent two crucial ingredients of the theory of glacial cycles. The role and operation of the global carbon cycle during glacial cycles will

be discussed below.

    Another potentially important positive feedback (a set of feedbacks) is related to global dust cycle. Paleoclimate reconstructions and modelling results show that the atmosphere was much dustier (typically by factor 2 globally) during the





ice ages (Kohfeld and Harrison, 2001; Albani et al., 2018). This fact is attributed to reduced vegetation cover, exposed continental shelf, and dust production due to glacial erosion processes (Mahowald et al., 2006). Radiative forcing of aeolian
dust strongly varies regionally, and its globally average magnitude remains uncertain (Bauer and Ganopolski, 2014). Apart from changing optical properties of the atmosphere, dust deposition strongly affects the surface albedo of snow and, therefore, the surface mass balance of ice sheets (Krinner et al., 2006; Ganopolski et al., 2010; Willeit and Ganopolski, 2018). In addition, a significant increase of aeolian dust deposition over the Southern Ocean through the iron fertilization effects led to the increase of net primary production of marine ecosystems and thus contributed to the glacial lowering of atmospheric $CO_2$ concentration
(Martin, 1990; Watson et al., 2000).

Another potentially important regional feedback is climate-vegetation feedback. Modelling results suggest that the biophysical effect of vegetation cover change during glacial time produced about 0.5-1°C of additional global cooling (Ganopolski 2003; Crucifix and Hewitt, 2005; O'ishi and Abe-Ouchi, 2013) and that this feedback amplified initial cooling caused by orbital forcing over the northern continents during glacial inceptions (e.g. de Noblet et al., 1996; Calov et al., 2005b).
At the same time, the effect of terrestrial biosphere changes on atmospheric $CO_2$ concentration due to the shrinking of terrestrial carbon pool during glacial times likely worked in the opposite direction, i.e., produces a negative feedback (see discussion below).

**5.3 Time scales of the Earth system response and the nature of 100 kyr cyclicity**

As shown in the previous section, the dominant periodicity close to 100 kyr of the glacial cycles during the last million years
originates in our models from the phase locking of glacial cycles to the corresponding eccentricity cycle. This mechanism requires that the typical duration of glacial cycles forced by obliquity and precession components of orbital forcing should be an order of 100 kyr. However, conceptual models derived from CLIMBER-2 do not have such a long internal time scale. The response time of Northern Hemisphere ice sheets to orbital forcing (30 kyr) used in Model 3 and adopted from Calov and Ganopolski (2005) is much shorter than 100 kyr. However, as shown above (see also Fig. 7), such long glacial cycles can arise
if the time needed to reach a critical ice sheet volume is much larger than one precession cycle.

Admittedly, even the existence of a 30 kyr time scale of the cryosphere response to orbital forcing is not trivial. A typical accumulation rate over the entire Northern Hemisphere ice sheets during glacial conditions is about 0.1-0.3 Sv[1] (Ganopolski et al., 2010) which is consistent with the results of LGM simulations with GCMs. The total Northern Hemisphere surface ablation and the solid ice discharge into the ocean (calving flux) vary strongly in time but are generally around 0.1 Sv each
(Ganopolski et al., 2010). Thus, it would be natural to consider 0.1 Sv as a typical value for the total ice sheet mass disbalance. Such an estimate corresponds to a complete growth/melt of the late Quaternary ice sheet in 10,000 years, which is just one-half of the precession cycle. In reality, such a rate of change is only achieved during glacial terminations, while for the rest of

---

[1] Here, the oceanographic unit of volume flux, Sverdrup, is used. $1Sv=10^6$ m³/s is approximately equivalent to 10 meters global sea level rise in 1000 years.



glacial cycles, the typical rate of global ice volume change was several times smaller, which implies that the positive and negative components of the ice sheet mass balance are close to each other by absolute values most of the time. This can be 540 explained by the fact that, apart from the positive albedo- and elevation feedbacks, there is strong negative feedback associated with ice sheets southward expansion. Model simulations show that the positive component of mass balance (accumulation) is roughly proportional to the size of ice sheets, whereas ablation increases strongly nonlinearly with the southward expansion of ice sheets margins. This negative feedback prevents North American and Eurasian ice sheets from spreading into low latitudes. As a result, reaching the equilibrium state takes much longer than one would expect from a simple scale analysis. 545 Only under extremely strong orbital forcing (during periods of high eccentricity) can the disbalance between the components of surface mass balance be much larger, and the rate of volume changes increases substantially. This can explain a few "short" glacial cycles at ca. 600 and 220 ka, which Model 3 cannot simulate by its design.

As shown in experiments with Model 3, phase locking of glacial cycles to 100 kyr periodicity of eccentricity originates from the highest likelihood of reaching the critical ice volume during periods of low eccentricity. This is explained by the fact 550 that, while high eccentricity facilitates ice sheets growth during glacial inceptions, the critical ice volume can be reached only if the system stays sufficiently long in the attraction domain of the glacial state, which occurs during periods of low eccentricity. In the case of realistic orbital forcing, the largest probability of reaching the critical ice volume is when a relatively weak maximum of precession component of insolation coincides with a minimum of obliquity. Thus, according to GMT, the timing and periodicity of glacial cycles of the late Quaternary are set by the shortest (100 kyr) eccentricity cycles. The appearance of 555 the eccentricity period in the spectra of glacial cycles originates from the phase-locking rather than the forcing of glacial cycles by eccentricity. This concept eliminates the main criticism of "the eccentricity myth" (Maslin and Brierley, 2015) based on the fact that the direct impact of eccentricity on insolation in terms of the global mean value is very small (about 0.5 W/m$^2$) to affect glacial cycles. According to GMT, eccentricity affects glacial cycles indirectly through its amplitude modulation of the precession component of insolation, which results in the variations of maximum boreal summer insolation at the top of the 560 atmosphere with an amplitude of more than 50 W/m$^2$. It is important to note that the term "phase locking" has here a different meaning than in the conceptual models where glacial cycles represent self-sustained oscillations like in the minimal conceptual model MiM described above or the model by Tziperman et al. (2006).

Obliquity plays a role in setting the timing of glacial cycles, but as was shown in Ganopolski and Calov (2011), the dominant 100 kyr cyclicity is simulated by CLIMBER-2 even if the obliquity component is artificially eliminated from orbital 565 forcing. The main difference in simulations with and without the obliquity component (see Fig. 3d in Ganopolski and Calov, 2010) is that elimination of obliquity leads to an increase of energy in the 405 kyr band, which is another eccentricity period. A qualitatively similar effect is seen in Model 3 results (Appendix A5, Fig A6), but the magnitude of the 405 kyr peak is strongly parameter-dependent. Thus, although obliquity affects the duration of individual glacial cycles, it plays no role in setting the 100 kyr periodicity. In other words, 100 kyr periodicity originates not from the fact that 100 kyr is close to two and 570 a half of obliquity periods but, to the contrary, glacial cycles typically lasted 2 or 3 obliquity cycles because $2.5 \cdot 41 \sim 100$ kyr.





It is important to stress that, although a sharp peak at 100 kyr in the frequency spectra of the Late Quaternary climate variability is successfully reproduced by several physically based models and is robust within a broad range of the parameters of conceptual models, this is a rather peculiar regime of operation of the Earth system. Such regime  is only possible for a "proper" combinations of orbital forcing and key boundary conditions: position of continents, regolith distribution, $CO_2$ level,

etc. This is why this regime was established only about 1 Ma, and it is likely that it will change to another regime in the future due to the natural evolution of the Earth system.

### 5.4 Multiple equilibrium states of the climate-cryosphere system and escape from the glacial trap

As shown in the previous section, the conceptual model based on multiple equilibrium states successfully simulates Quaternary glacial cycles. The idea that the climate-cryosphere has several equilibrium states has a long history (e.g., Budyko, 1972;

Weertman, 1976; Benzi et al., 1982). Multiple equilibrium states of the climate-cryosphere system have also been found in several models that explicitly included ice sheet components (e.g., Pollard and DeConto, 2005; Calov and Ganopolski, 2005; Robinson et al., 2012; Abe-Ouchi et al., 2013). These studies show that different ice sheets have very different stability diagrams in the phase space of orbital forcing, temperature, or $CO_2$. In the simplest case, the stability diagram of an ice sheet in the phase space of orbital forcing (maximum summer insolation at 65°N) is represented by two branches (glacial and

interglacial) as in Model 3 (Fig. 11a). It is also common for illustrating purposes to depict such a bi-stable system in the form of double-well potential (Fig. 11b). Each well represents one of the equilibrium states, and the depth of the wells represents the stability of each state. Since the shape of such a diagram depends on orbital forcing itself, the diagram in Fig. 11b shows potential for average orbital forcing, i.e., zero forcing anomaly.

If the system is in the interglacial state, it will remain in this state as long as the orbital forcing stays above the critical

threshold value $F_{crt}$. This value depends on $CO_2$ concentration (Calov and Ganopolski, 2005; Archer and Ganopolski, 2005). In Ganopolski et al. (2016), this dependence on $CO_2$ has been systematically studied, and a simple logarithmic relationship between the critical value of insolation and $CO_2$ concentration has been found:

$$F_{crt} = \alpha \ln \frac{CO_2}{280} + \beta,$$

where $\alpha$ and $\beta$ are constants. In the phase space of orbital forcing, glacial inception represents a bifurcation transition from

the glacial to the interglacial state (Calov et al., 2005a). However, this transition is "abrupt" only in the phase space. In reality, because of the very long response time of the climate-cryosphere system to orbital forcing, the system's trajectory differs significantly from the stability diagram shown in Fig. 11a. However, when the system enters the glacial cycle, it tends to stay longer in the domain of the attraction of glacial equilibria even though insolation is below its critical value $F_{crt}$ in average significantly less than 50% of the time. Fig 11a illustrates this "irreversibility" of glacial inception. While such behaviour is

entirely consistent with the paleoclimate reconstructions, the principal question arises: if the glacial equilibrium is so stable, why does the Earth system rapidly move into the interglacial state instead of oscillating around the glacial equilibrium state?. In other words, the question is how does the Earth system escape its "glacial trap"? The stability diagram shown in Fig. 11a





cannot explain such behaviour because, as shown above, based on this stability diagram, Model 3 cannot simulate deglaciation

without introducing an additional deglaciation regime shown in Fig. 11a by the red line. In Fig 11b this deglaciation trajectory is shown as the transition through the potential barrier separating two wells. To some extent, this process is analogous to the quantum tunnel effect. This mechanism of escape from the glacial trap represents one of the key elements of the GMT and is discussed below.

## 5.5 Glacial terminations: the domino effect

The most serious challenge to the classical Milankovitch theory is the strong asymmetry of the glacial cycles during the late

Quaternary. The domination of such long glacial cycles requires that a relatively small ice sheet can survive periods of rather high summer insolation while larger ice sheets vanish completely even after modest insolation rise as it happened, for example, at the onset of MIS11 ca. 430 ka. The instability of the ice sheets after reaching a "critical size" has been postulated in P98 and is also essential for conceptual models described in this paper. The question is how to explain this instability and what "critical size" means. The analogy with the mechanical "domino effect" is helpful in answering these questions.

The domino is an example of a mechanical system with different equilibrium states and can respond to an external forcing both linearly and strongly nonlinearly (the "domino effect") depending on one parameter value – the distance between the dominos. Let us consider a pendulum hitting the row of dominos, as shown in Fig. 12. If this distance between dominoes is even slightly larger than the height of dominoes, the pendulum will only knock down the dominoes within its reach. In this case, the number of dominoes that fall is proportional to the amplitude of pendulum oscillations. However, if the distance

between dominoes is only slightly smaller than their height (this is the situation used in demonstrations of the "domino effect"), then all dominoes in the row will fall irrespectively of the amplitude of pendulum oscillations.

The unstoppable retreat of the Northern Hemisphere ice sheets, which occurs in response to insolation rise and which final result (complete deglaciation) does not depend on the magnitude of insolation rise, resembles the "domino effect." In this case, the analogue for the distance between the dominoes is the "critical size" of ice sheets. Analysis of CLIMBER-2 model

simulations suggests that the concept of "critical size" applies primarily to the North American ice sheet since the response of the smaller Eurasian ice sheet to orbital forcing is more linear. Other opinions exist; for example, Paillard and Parrenin (2004) attributed "critical size" to the Antarctic ice sheet.

What causes the domino effect in the case of supercritical ice sheets? It is likely that at least several processes and mechanisms must be considered to explain why a large North American ice sheet is so sensitive to insolation rise and can

vanish at a time scale about or even shorter than 10,000 years.

1. Large North American ice sheet spreads over the vast areas covered by unconsolidated sediments (Great Lakes region, the western prairies of the US and Canada), where the ice sheet is thinner and flatter than over areas with exposed rocks. This is explained by a much larger basal sliding velocity over unconsolidated water-saturated sediments than bare rocks (e.g., Licciardy et al., 1998). This implies that the slope of a large ice sheet near its margins is less steep, and as a result, the ablation





area is larger (Fig. 13a), which explains a higher sensitivity of the surface mass balance of such ice sheet to insolation and $CO_2$ rise.

2. The expansion of ice sheets over the areas covered by a thick terrestrial sediment layer leads to the production of a large amount of glaciogenic dust, which originates from the sediments transported beneath ice sheets across their margins (Mahowald et al., 2006; Ganopolski et al., 2010). A fraction of these sediments become airborne, and while this glaciogenic dust precipitates over the ice sheet, it significantly reduces surface reflectivity as the albedo of snow is very sensitive to even a tiny concentration of impurities. (Warren and Wiscombe, 1980; Dang et al., 2015). This, in turn, has a significant impact on the surface mass balance of ice sheets (Krinner et al., 2006; Willeit and Ganopolski, 2018) and causes the ice sheets to retreat earlier (Fig. 13b).

3. The weight of a mature ice sheet causes significant bedrock depression, roughly equal to 1/3 of the ice sheet thickness, i.e. reaches more than 1 km in the centre of continental ice sheets. A typical time scale of the bedrock relaxation towards its unperturbed (approximately modern) state after the removal of the ice sheet is about 5000 years, which is small compared to the time scale of a glacial cycle (100 kyr) but is comparable to the time scale of glacial terminations (10 kyr).

In the case of rapid retreat of ice sheets, delayed bedrock rebound facilitates surface melt. This is explained by the fact that, for the same thickness of ice sheet, the surface elevation in the ablation zone is lower in the case of delayed relaxation than it would be in the case of instantaneous bedrock adjustment (Fig. 13c). The role of delayed bedrock adjustment in the shaping of glacial cycles has been demonstrated already in Oerlemans (1980) and Pollard (1983). Simulations with a 3-D ice sheet model (Abe-Ouchi et al., 2013) confirmed that delayed bedrock adjustment is important for the complete deglaciations of the Northern Hemisphere at the end of each glacial cycle.

4. Another mechanism which can contribute to fast deglaciation is the so-called "marine ice sheet instability" (Weertman, 1974). This mechanism requires that the base of the ice sheet is located below sea level and the retrograde slope of bedrock (the elevation of bedrock decreases inland). This mechanism can cause a relatively rapid (millennial time scale) disintegration of a significant fraction of the ice sheet without significant surface mass loss. It was proposed that the Barents ice sheet disintegrated through this mechanism around Bolling warm event (Brendryen et al., 2020).

5. The idea that the formation of proglacial lakes south of the ice sheets during their final retreat plays an important role during glacial terminations (Andrews, 1973) has also been first tested in Pollard (1983). Large proglacial lakes are formed in depressions resulting from delayed bedrock relaxation in the process of fast ice sheet retreat. These depressions areas are filled by the water from melting ice sheets, and their level can be well above sea level. The formation of the proglacial lakes leads to another mechanism of ice sheet instability which can be considered as the freshwater analogue of marine ice shelf instability (Quiquet et al., 2021; Hinck et al., 2022). The appearance of proglacial lakes causes a significant increase in the ice flux through the grounding line into the lakes (Fig. 13d). This ice then rapidly melts because it has a low elevation.

6. Rapid $CO_2$ rise during glacial terminations plays in tandem with the insolation rise during glacial terminations. The modelling study suggests that both factors play a comparable role in the ice sheet mass loss (Heinemann et al., 2014; Gregorie et al., 2015). The rapid melting of ice sheets during terminations results in large freshwater flux into the North Atlantic (0.1-



0.2 Sv) is sufficient to cause a prolonged shutdown of the AMOC, which in turn additionally contributes to the deglacial $CO_2$
rise and its overshoot at the end of deglaciation (Ganopolski and Brovkin, 2017). At the same time, AMOC shutdown during
deglaciation causes anomalous cooling over Europe and slow-down deglaciation (negative feedback). Additionally, subsurface
warming induced by the AMOC shutdown can destabilize ice shelves in the North Atlantic realm (see below). The net effect
of AMOC changes on deglaciation still has to be evaluated with adequate Earth system models.

     All mechanisms discussed above are likely important for the "domino effect". Each of the mechanisms discussed above
should not necessarily be "catastrophic", but  when they work together, the combined effect can cause rapid disintegration of
large continental ice sheets. It is also important to stress that none of these mechanisms are specific for the 100 kyr world – all
of them also operated during 41 kyr world and MPT. According to GMT, the difference between 41 kyr world and 100 kyr
world is in the critical size of ice sheets after exceeding of which, the mechanisms discussed above lead to rapid deglaciation
of the Northern Hemisphere.

**5.6 The role of climate-carbon cycle feedbacks**

It is generally recognized that $CO_2$ plays a vital role in the dynamics of glacial cycles. It is also important to distinguish between
the "effect" of glacial $CO_2$ on climate variability and the role of carbon cycle feedback during glacial cycles. On the geological
time scales, the average $CO_2$ concentration is controlled by the interplay between volcanic outgassing and weathering rate. In
turn, this concentration controls the amplitude of glacial cycles (Berger et al., 1999; Ganopolski and Calov, 2011; Abe-Ouchi
et al., 2013), and for a sufficiently high $CO_2$ concentration, glacial cycles are not possible for a realistic range of the Earth's
orbital parameters. This is why gradual lowering of $CO_2$ concentration below a certain threshold was one of the preconditions
for the onset of Quaternary glacial cycles (Willeit et al., 2015). After $CO_2$ concentration dropped below this threshold and
regular glacial cycles began, climate-carbon cycle feedback amplified glacial cycles, especially their 100 kyr component and
also globalized their impact on climate.

The quest for the mechanisms of glacial-interglacial $CO_2$ variability discovered thanks to the Antarctic ice cores (Petit et
al., 1999; Lüthi et al., 2008) attracted significant attention during the past two decades. However, the mechanisms of this
variability are still not fully understood. Results of numerous modelling studies strongly indicate that there is no single
mechanism that can explain most of the observed glacial-interglacial $CO_2$ changes of about 100 ppm, and it becomes
increasingly clear that at least several processes should be involved (e.g. Archer et al., 2000; Kohfeld and Ridgwell, 2009).
Moreover, it is likely that the role of geochemical (inorganic) and biogeochemical processes in glacial-interglacial $CO_2$
variability are of comparable importance (Galbraith and de Lavergne, 2019). It is much more certain that glacial $CO_2$ lowering
can only be explained by the ocean carbon uptake since the terrestrial carbon pool was significantly depleted during the glacial
time due to the shrinking of the vegetation cover (Brovkin et al., 2012; Jeltsch-Thommes et al., 2019). It should be noted that,
apart from the redistribution of carbon between atmospheric, land and ocean pools,  the disbalance between volcanic outgassing
and weathering rate may also play a role in $CO_2$ variability at the orbital and longer time scales.





It is known that $CO_2$ concentration and global ice volume during the late Quaternary are highly correlated – the correlation coefficient between $CO_2$ and ice volume is above 0.7 for the last 800 kyr. However, the high correlation alone says nothing about causal relationships. Modelling studies suggest that the direct response of the carbon cycle to orbital forcing is weak. This is also supported by the fact that obliquity and precession components are relatively weak in $CO_2$ records of the Late

Quaternary. At the same time, modelling results demonstrate that strongly asymmetric glacial cycles with the dominant 100 kyr periodicity can be simulated even with a constant $CO_2$ concentration if this concentration is sufficiently low (Ganopolski and Calov, 2011; Abe-Ouchi et al., 2013). This is consistent with the concept that 100 kyr cyclicity originates from a nonlinear response of the climate-cryosphere system to orbital forcing and that the climate-carbon cycle feedback amplifies the 100 kyr component of glacial cycles (Ganopolski and Brovkin, 2017). However, while the influence of $CO_2$ through climate on global

ice volume is rather straightforward, the influence of ice sheets on $CO_2$ concentration is far from trivial. The most direct effect of ice sheets growth on $CO_2$ is through the decrease of the ocean volume, which leads to the effect opposite to the observed $CO_2$ changes. Indeed, the decrease of the ocean volume by ca 3% at the LGM would cause $CO_2$ to rise by about 10 ppm. In addition, the growth of ice sheets reduces forest area in the boreal zone, which also should cause a $CO_2$ rise. The direct effect of ice sheets on the carbon cycle, which can contribute to the glacial $CO_2$ drawdown, is regional cooling due to higher albedo

of ice sheets. This cooling leads to a lowering of ocean temperature, mainly in the vicinity of the ice sheets, and contributes to $CO_2$ decrease through the solubility effect. However, changes in ice sheets area primarily affect the climate of the Northern Hemisphere, and even the total solubility effect (including lowering of $CO_2$ and other GHGs concentrations), explains not more than 30 ppm of glacial $CO_2$ decrease (Brovkin et al., 2007; Kohfeld and Ridgwell, 2009).

In addition to global ocean volume change, ice sheets can affect $CO_2$ through the global sea level change. First, this leads

to the change in ocean alkalinity through the dissolution or regrowth of shallow-water carbonates, particularly in the form of corrals (Ridgwell et al., 2003). Higher alkalinity, which results from the sea level drop, increases ocean capability to absorb atmospheric $CO_2$ and thus contributes to glacial $CO_2$ drawdown. Second, exposed shelves may represent significant additional sources of the aeolian dust, and enhanced dust deposition can intensify marine biological productivity in the Southern Ocean, which is iron-limited (Martin, 1990; Watson et al., 2000; Yamamoto et al., 2019). Third, due to the peculiar oceanic

hypsometry, even a relatively small initial sea level drop leads to a significant increase in land area. This not only contributes to additional global cooling but also leads to an expansion of vegetation cover and thus serves as the carbon uptake. The latter can work only during the initial phase of glacial cycles because, under the full glacial conditions, global decline in terrestrial biomass due to ice sheets expansion and $CO_2$ lowering overwhelms the effect of land area increase. In this way, there are at least several mechanisms through which ice sheets can cause changes in $CO_2$ coeval with ice volume changes.

This initial response of atmospheric $CO_2$ to ice sheets evolution is amplified by several climate-carbon cycle feedbacks: 1) enhanced ocean carbon uptake due to cooling, i.e., solubility effect, additional to the effect caused by ice sheets growth; 2) the effect on ocean carbon storage caused by changes in the ocean circulation, stratification and sea ice cover leading to changes (reduction) in the deep ocean ventilation; 3) change in the remineralization depth caused by ocean thermocline temperature lowering which affects organic carbon flux into the deep ocean. Although the operation of the global carbon cycle during





glacial times is still not fully understood, it is likely that the direct effect of ice sheets on $CO_2$ and the effect of its amplification via climate-carbon cycle feedbacks are of comparable importance.

## 5.8 Glacial cycles of the early Quaternary (41 kyr world)

Prior to the MPT (ca 1.2-0.8 Ma), glacial cycles had a smaller magnitude (50% when compared to the late Quaternary in terms of benthic $\delta^{18}O$), and the dominant periodicity was 41 kyr, which is the periodicity of obliquity with very little energy in 100

kyr and precession bands (Fig. 2). According to the GMT, a shorter periodicity and smaller magnitude of early Quaternary glacial cycles are explained by the fact that the critical size of ice sheets, which makes the Earth system  prone to deglaciation, was smaller before the MPT than after. The most likely explanation for that (see also the next section) is the gradual removal of terrestrial sediments from North America by glacial erosion (Clark and Pollard, 1998). The thick terrestrial sediments make ice sheets more mobile since "temperate" ice (ice at the pressure melting point) moves much faster over sediments than over

bare rocks. Thus, the presence of thick sediments allows ice sheets to spread faster and makes them thinner, which also explains a larger sensitivity of ice sheets mass balance to orbital forcing. In addition, the spreading of ice sheets over terrestrial sediment led to the production of a large amount of glaciogenic dust, a part which was then deposited over ice sheets, making them darker and thus increasing surface ablation (Ganopolski and Calov, 2011). As a result, the early Quaternary ice sheets were able to reach their critical size during just one obliquity cycle. Since glacial cycles were much shorter, these cycles look more

symmetric than the Late Quaternary cycles, but this does not mean that the response of the Earth system to orbital forcing was linear, and it is likely that all elements of the "domino effect" also operated during glacial terminations of the early Quaternary.

While the dominance of the obliquity component in ice volume variations during the early Quaternary is not surprising and is reproduced in model simulations (Ganopolski and Calov, 2011; Willeit et al., 2019), a strong precession component is also clearly present in the spectra of simulated ice volume. At the same time the precession component is essentially absent during

early Quaternary in many paleoclimate reconstructions, including the canonical LR04 benthic stack. This mismatch is often considered another unsolved problem ("mystery") of the Milankovitch theory (Raymo and Nisancioglu, 2003). A number of explanations for the lack of precession components prior to MPT have been proposed, including a possibility for a mutual compensation of precession components originating from southern and northern ice sheets evolution (Raymo et al., 2006). However, in recent years, precession signal during the early Quaternary started to emerge in paleoclimate records (e.g. Schakun

et al., 2016; Liautaud et al., 2020; Barker et al., 2022), and there may be an inherent issue with absolute dating of old paleoclimate records. Appendix A6 illustrates how a simple "orbital tuning" can nearly completely eliminate the precession signal from the ice volume record. However, there is another, often overlooked aspect of 41-kyr "mystery": it is generally assumed that the entire Quaternary records are dominated by ice volume variability. While about 2/3 of benthic $\delta^{18}O$ variability (or 80% in spectral power) is attributed to global ice volume change during the late Quaternary, results of model simulations

(Willeit et al., 2019) show that the relative contribution of ice volume to benthic $\delta^{18}O$ was only about 1/3 (or only 20% in the frequency spectra) before the MPT. This is consistent with Elderfield et al. (2012), which also found that the relative


contribution of deep ocean temperature to $\delta^{18}$O prior to MPT was significantly higher than 50%. This would change the "41 kyr mystery" from the challenge of explaining the absence of precession component in global ice volume (which is indeed hard to explain) to the question of why the deep ocean temperature does not contain much precession variability. The latter is
a different problem since the deep ocean temperature is controlled by different factors than the ice volume, primarily by $CO_2$ concentration. Since $CO_2$ concentration over the past 800 kyr contains very little precession variability (e.g. Petit et al., 1999), it would be natural to assume that this was the case also during the early Quaternary. Thus the "41 kyr mystery" cannot be considered a fundamental problem of the Milankovitch theory.

**5.9 Three Quaternary regime changes: PPT, MPT and MBT**

During the past three million years, climate variability at orbital time scales revealed three distinct changes in the mode of operation. The first one, referred to as the Pliocene-Pleistocene Transition (PPT), is the onset of regular glacial cycles with the dominant 41-kyr periodicity, which happened between 2.8 and 2.7 Ma, i.e., significantly earlier than the "official" stratigraphic Pliocene-Pleistocene border (2.58 Ma). The second transition, the Mid-Pleistocene Transition (MPT) from 41 to 100 kyr world occurred between 1.2 and 0.8 Ma. The third transition, clearly seen only in some paleoclimate records (such as $\delta^{18}$O and $CO_2$)
is referred to as mid-Brunhes Transition[2] and manifests itself in the differences between interglacials before and after MIS11.

The onset of Quaternary glacial cycles has been preconditioned by the general Cenozoic cooling trend (Raymo and Ruddiman, 1992) caused by decreasing $CO_2$ concentration and the poleward advance of North America during the Cenozoic epoch (Daradich et al., 2016). Several additional proximal causes, including the closing, opening or deepening of different oceanic gateways, establishing of North Pacific stratification, etc., have been proposed to explain PPT (e.g., Ruddiman and
Raymo, 1988; Cane and Molnar, 2001; Haug and Tiedemann, 1998; Haug et al., 2005) but the role of these events in the onset of Quaternary glacial cycles is not clear. At the same time, it has been shown (Willeit et al., 2015) that even a gradual downward $CO_2$ trend alone is sufficient to cause the transition from essentially ice-free Northern Hemisphere to glacial cycles of medium amplitude after the PPT. The critical value of $CO_2$ concentration of 350-400 ppm below which regular glaciations in the Northern Hemisphere can begin (Ganopolski et al., 2016) is consistent with the recent $CO_2$ reconstructions during the PPT
(Martínez-Botí et al., 2015).

In the 1.5 million years which followed the PPT, the magnitude of glacial cycles increased gradually, but the dominant periodicity (41 kyr) remained unchanged. However, between 1.2 and 0.8 Ma, climate variability recorded by $\delta^{18}$O changed dramatically from relatively symmetric cycles of the 41-kyr world to strongly asymmetric 100 kyr cycles. The magnitude of glacial cycles expressed in the standard deviation of benthic $\delta^{18}$O doubled across the MPT (Lisiecki and Raymo, 2007).

The MPT has been attributed by Clark and Pollard (1998) to a gradual evolution of terrestrial sediment cover over northern continents (the so-called "regolith hypothesis"). This hypothesis is based on the empirical fact that at present large areas of

---

[2] Since no real "event" is associated with this transition, the term MBT seems to be more appropriate than the traditional term "Mid-Brunhes Event".



northern North America and Eurasia are characterized by a thin layer of terrestrial sediments or exposed crystalline rocks, while the rest of these continents are covered by kilometre-thick terrestrial sediments. Since the areas of thin sediments/exposed bedrocks coincide with the areas covered by ice sheets most of the time during Quaternary, it is natural to assume that, prior

to Quaternary, these areas were also covered by a thick sediment layer which was then gradually removed by glacial erosion processes. In turn, the underlying surface type (crystalline rocks or unconsolidated terrestrial sediments) strongly affects ice sheet dynamics as ice can slide much faster over sediments (compared to rocks) when the temperature at the base of ice sheets reaches the pressure melting point. This can explain why ice sheets of the early Quaternary were thinner and thus more susceptible to insolation changes. As a result, ice sheets during the Early Quaternary responded to orbital forcing in a more

linear manner without any appreciable contribution from the eccentricity cycles. The appearance of large areas of exposed crystalline rocks about 1 million years ago led to the formation of thick and slower spreading ice sheets with a narrower ablation zone. The latter helped ice sheets to survive several insolation maxima before they spread deep enough into the sediment-covered areas, which made them prone to a rapid collapse during insolation rise. In Ganopolski and Calov (2011) we proposed an additional mechanism that might contribute to the MPT and which is also related to the removal of the regolith

layer from the northern continents. This mechanism is based on the fact that, when ice sheets spread over the areas covered by terrestrial sediments, they produce a significant amount of glaciogenic dust, which, in turn, affects surface albedo and ablation (see Section 5.5). While this mechanism operated through the entire glacial cycle before the MPT as northern continents were covered by terrestrial sediments, ice sheets spread over the sediment-covered areas only after they approached their critical size after the MPT. Both these mechanisms were incorporated into the CLIMBER-2 model and contributed to the realistic

simulation of MPT (Willeit et al., 2019).

It has also been shown by Willeit et al. (2019) that even a gradual expansion of the sediments-free areas from zero to the present one over the past 1500 kyr can explain a relatively rapid transition (over several hundred thousand years) from the 41-kyr to 100 kyr regime. In other words, it has been shown that there is a critical threshold for the exposed rocks area, crossing of which leads to the transition from 41 kyr cyclicity to 100 kyr cyclicity. It has to be noted that a gradual reduction of $CO_2$

concentration during Quaternary is considered by some authors (e.g., Berger et al. 1999) as a possible cause of MPT, according to our result, is not an alternative mechanism of MPT but rather an important precondition for the onset of long glacial cycles, since for high $CO_2$ concentrations, only relatively weak and short glacial cycles are possible even for the present-day regolith distribution, (Fig. S9 in Willeit et al., 2019). Thus, the onset of the 100 kyr world has been preconditioned both by $CO_2$ decline and regolith removal. However, the timing of the MPT was likely set by the regolith removal since there is no strong evidence

for the decline of $CO_2$ level directly prior to the MPT (Hönisch et al., 2009; Chalk et al., 2017; Yamamoto et al., 2022).

Apart from regolith removal and $CO_2$ lowering, several other mechanisms have been proposed for explaining the MPT. For example, it has been speculated that this regime change of glacial climate variability may have been related to changes in the operation of the global carbon cycle and ocean circulation (e.g., Chalk et al., 2017; Farmer et al., 2019; Hasenfratz et al., 2019). However, it is also possible that observed changes in any paleoclimate proxy across the MPT can result from the

increase in the magnitude of glacial cycles (i.e., ice volume variations) and not be the cause of this change. It is also likely that





the larger magnitude of sea level and temperature fluctuations after the MPT can also explain the expansion of the marine-based Antarctic ice sheet (Ford and Raymo, 2020; Sutter et al., 2019). Only results of Earth system model simulations can confirm or reject a possible contribution of all these mechanisms to the MPT.

The last transition – MBT – is less clearly defined, and the number of long glacial cycles before and after the MBT is too small to obtain robust statistics. The most apparent change across the MBT is seen in $\delta^{18}O$ and $CO_2$ during interglacials. These differences are consistent with the idea that pre-MBT glacial terminations were "incomplete," and that a certain amount of "glacial" ice (ca. 10-20 meters in sea level equivalent) remained in ice sheets/ice caps in the Northern Hemisphere outside of Greenland during pre-MBT interglacials. This amount of ice, in combination with lower interglacial $CO_2$ concentration, explains the difference in $\delta^{18}O$ of about 0.3‰ between Holocene and corresponding values during MIS15 and MIS 19 interglacials. Interglacials MIS 13 and 17 were even colder, and a significant amount of ice (20 to 40 in msl equivalent) would be required to explain observed $\delta^{18}O$ differences.

The cause of MBT remains unclear. Although orbital forcing during the intervals 800-400 ka and 400-0 ka was not identical, there is no obvious reason why pre-MBT terminations were incomplete. One possible explanation for MBT is that glacial cycles of the late Quaternary were affected by glacial erosion not only through the gradual regolith removal but also through the curving of straights, fiords, and bays. It is believed that such important features of modern geography as Hudson Bay and Hudson Straight were formed by glacial erosion very recently (by geological time scale). At least pronounced Heinrich-type events were only observed since MIS12 (Hodell et al., 2008). This type of landscape evolution would lead to the development of fast-moving ice streams over North America, reducing the height of Laurentide ice sheets and facilitating complete deglaciations of the northern continents during post-MBT glacial terminations.

## 5.10 Forced events versus internal oscillations

As discussed above, formally, there are two different types of conceptual models that can simulate 100 kyr cycles. In the first type, glacial cycles represent self-sustained oscillations with a period close to 100 kyr and which do not require even the existence of orbital forcing. In the second type of conceptual models, glacial cycles represent a strongly nonlinear response of the Earth system to orbital forcing. In this case, 100 kyr periodicity is related to eccentricity, and glacial cycles cannot exist without orbital forcing. Note that CLIMBER-2 and our conceptual models (Model 2 and 3) belong to the second type. Moreover, so far, none of the Earth system models was able to simulate 100 kyr long self-sustained internal oscillations.

To decide which one of these two concepts is correct solely based on the analysis of paleoclimate data is problematic because orbital forcing is always present, and thus it is impossible to conclude whether it is necessary for driving glacial cycles. However, there is one feature of modelled glacial cycles that can be used to distinguish between these two mechanisms. This feature is related to the role of system memory. For the self-sustained oscillations, such memory is crucial, and it is not possible for the model's prognostic variables (e.g., ice volume) to stay constant for a finite time. This is akin to a mechanic clock – it cannot stop for a while and then resume its work without external influence. On the contrary, in models with forced oscillations (like Model 3), constant (zero) ice volume can stay constant for any duration of time until orbital forcing crosses the glaciation





threshold. What does paleoclimate data tell us in this respect? The paleoclimate records suggest that at least during three recent
interglacials (Holocene, Eemian, and MIS 11), sea level, $CO_2$, and other climate characteristics remained nearly constant over
a rather long time (ca. 10 kyr during Holocene and Eemian, and probably about 20 kyr during MIS11). Moreover, under a
weak orbital forcing, the Earth system can stay in the interglacial state even longer. According to model simulations (e.g.,
Loutre and Berger, 2000; Ganopolski et al., 2016) at present, the Earth system is in an unusually long interglacial, which can
last naturally (without any anthropogenic influence) for as long as another 50,000 years (i.e., more than two precession cycles).
Such long interglacial never occurred during the previous million years but can happen several times during the next million
years (Talento and Ganopolski, 2021). The existence of such long interglacials is impossible to reconcile with the self-
oscillatory mechanism of glacial cycles.

### 5.11 Are glacial cycles "predictable"?

It has been argued (e.g., Crucifix 2013; Ashwin et al. 2018) that Late Quaternary glacial cycles are entirely or at least partly
random and therefore "unpredictable." Since predictability of future glacial cycles cannot be tested in the foreseeable future,
the term "predictability" can only be used in an operational sense, i.e., whether an accurate and robust hindcast of past glacial
cycles is possible. In practice, this can be formulated as follows: can one expect that a sufficiently "realistic" Earth system
model forced solely by orbital variations can reproduce not only the statistical characteristics of past glacial cycles (e.g.,
amplitude and typical periodicities) but also the correct timing of individual cycles. This would be highly unlikely if glacial
cycles are totally "unpredictable."

However, the results obtained with the process-based models of different complexity show that not only the statistics but
also the timing of the late Quaternary glacial cycles can be accurately reproduced when driving models by orbital forcing alone
(Pollard 1982; Berger et al. 1999; Ganopolski and Calov 2011; Abe-Ouchi et al., 2013; Willeit et al., 2019). The latter study
also demonstrates a weak dependence of the simulated glacial cycles on the initial conditions. These modelling results strongly
indicate that glacial cycles are at least quasi-deterministic and thus predictable in principle. This does not contradict the fact
that different model solutions can be obtained when using different initial conditions or model parameters. Model simulations
show that the timing of some terminations is more robust than others. For example, in Willeit et al. (2019), although the timing
of most glacial terminations was robust across the large ensemble of simulations, the last glacial termination occurred earlier
than in reality in some model runs.

### 5.12 Antarctic ice sheet

The previous discussion was mostly restricted by the evolution of the Northern Hemisphere ice sheets. Of course, such a
northern-centric approach is only applicable to the direct ice sheet response to the orbital forcing. The carbon cycle - climate
feedback which amplifies and globalizes this response is truly global, and, in particular, the Southern Ocean plays an important
role in driving glacial-interglacial $CO_2$ variations. Still, it would be natural to ask about the role of the Antarctic ice sheet (AIS)
in glacial cycles, which is by far the largest ice sheet at present.



According to the GMT, the evolution of AIS during glacial cycles represent a passive response to two factors driven from the north: sea level change and GHG concentration variations. Both factors – sea level drop and ocean cooling – led to the turning of the ice shelf into grounding ice. The current estimates of LGM contribution to the global ice volume are usually within the range of 5-15 msl (e.g. Briggs et al., 2014), which is, on average, about 10% of the "glacial excess" of the global

ice volume. This makes AIS a medium amplifier of the glacial cycles forced from the north. The impact of the Antarctic ice sheet expansion on the atmospheric $CO_2$ through the brine rejection process has been proposed (Bouttes et al., 2012), but this hypothesis still should be tested with realistic ESMs. This is, however, not a trivial task since such models have problems with simulation even of the present-day mode of Antarctic Bottom Water formation.

## 5.13 Simple rule to determine the timing of glacial terminations

The general concept of GMT allows to formulate a very simple rule for the timing of the late Quaternary glacial terminations: **Glacial terminations occur during periods of rising boreal summer insolation if the previous precession insolation maximum occurred during low eccentricity and was in antiphase with obliquity.**

To exclude a few false "positives," an additional constraint is required, namely, that glacial termination cannot occur in less than 60 kyr after the previous one. To construct a numerical algorithm for this rule, "low eccentricity" is defined as the

interval ± 25 kyr around each local eccentricity minimum. "Antiphase with obliquity" means here that the precession maximum occurs any time when obliquity is below its average value. The minimum rate of insolation growth sufficient for triggering deglaciation is set to a small value of 1 W/m$^2$ in 1000 years. As Fig. 15 shows, this very simple rule works rather well for the last 900 kyr. Nine of ten glacial maxima of the late Quaternary are predicted with the dating accuracy, i.e., 10 kyr. The only exception is the termination VIII at around 720 ka, which this rule does not predict. This termination is unusual as it began at

around obliquity minimum. It is noteworthy that neither CLIMBER-2 nor conceptual models (Model 2 and 3) have problems with simulating the correct timing of Termination VIII.

## 5.14 Short summary of GMT

According to the GMT, Quaternary glacial cycles represent the direct, strongly nonlinear response of the Earth system to variations of summer insolation in boreal latitudes of the Northern Hemisphere. This nonlinear response to orbital forcing is

strongly modified and amplified by several processes and feedbacks. The existence of Quaternary glacial cycles and especially the strongly asymmetric late Quaternary cycles are a peculiar, possibly, absolutely unique feature of the current Earth system "configuration". This explains significant difficulties in modelling and understanding Quaternary climate dynamics.

The onset of Quaternary glacial cycles was preconditioned by changes in the Earth's geography due to continental drift and by a gradual lowering of $CO_2$ concentration, which eventually brought the Earth system into the state when insolation variations

caused regular glaciations of the Northern Hemisphere. Before ca. 1 Ma, medium amplitude glacial cycles terminated each time when positive insolation anomalies due to precession and obliquity components of orbital forcing were in phase, which





led to the dominance of obliquity (41 kyr) periodicity. The precession component of orbital forcing also played an essential role in driving early Quaternary glacial cycles, even if this periodicity is not detected in global ice volume reconstructions.

Gradual removal of terrestrial sediments from the northern continents by glacial erosion processes brought the Earth system into a new regime of operation when the dominant periodicity of glacial cycles shifted from 41 to 100 kyr. This regime change is explained by the fact that a large area of exposed rocks in Northern America combined with a relatively low $CO_2$ allowed medium-sized ice sheets to survive periods of high summer insolation and reach a much larger size/volume than the ice sheets of early Quaternary. However, after reaching some critical state, a combination of several processes and feedbacks which are strongly dependent on the ice sheet size made such ice sheet unstable under boreal summer insolation rise. This led to rapid

deglaciations, which explains the strong asymmetry of the late Quaternary glacial cycles. Since the critical ice sheet size can most likely be reached during periods of low eccentricity when one "positive" precession cycle is compensated by a "negative" obliquity cycle, the glacial termination occurs during or after periods of low eccentricity. As a result, glacial cycles became phase-locked to short (100 kyr) eccentricity cycles, and this explains a sharp 100 kyr in the frequency spectra of the late Quaternary glacial cycles.

**6. Discussion and conclusions**

The central premise of the Milankovitch theory, namely that boreal summer insolation variations are the principal driver of Quaternary glacial cycles, has been supported first by analysis of paleoclimate records which reveal the presence of all expected astronomical periodicities and then by modelling results which confirmed that these variations could cause waning and waxing of the continental ice sheets in the Northern Hemisphere. Thus, the Milankovitch theory in its original form has been confirmed,

but paleoclimate records also revealed a number of important facts that were not known during Milankovitch's life and which his theory was not able to explain. The first one is that the late Quaternary is dominated by the strongly asymmetric glacial cycles with the dominant 100-kyr cyclicity. The explanation for these cycles has attracted significant attention and continues to be regarded by some researchers as a challenge. In fact, the temporal dynamics of the late Quaternary glacial cycles, including the timing of glacial terminations and frequency spectra of climate variability, have already been successfully

reproduced with a number of climate-ice models of different complexity ranging from the simple 1-D models to the comprehensive Earth system model. In these model simulations, the dominant peak at 100 kyr periodicity in the frequency spectrum of ice volume is explained by the phase locking of long glacial cycles of the late Quaternary to 100 kyr eccentricity cycles. The eccentricity, in this case, serves as a pacemaker for the precession- and obliquity-driven glacial cycles and the minuscule changes in global annual mean insolation associated with eccentricity cycles play no role here. The real forcing of

glacial cycles is large (50 -100 W/m$^2$) changes in boreal summer insolation with precession and obliquity periods.

    The idea that 100 kyr cyclicity originates from some strongly nonlinear response of the Earth system to the amplitude modulation of the precessional component of orbital forcing by the eccentricity, of course, is not new. However, it is important to emphasize that the late Quaternary glacial cycles cannot be explained simply by a nonlinear response because an arbitrary nonlinear response to orbital forcing should contain all eccentricity periodicities (95, 124, 400 kyr) as in the case of the Imbrie



and Imbrie model, for example. In reality, only a single, sharp peak at ca. 100 kyr is observed in the late Quaternary ice volume frequency spectrum. Thus, explaining paleoclimate records requires a very special type of nonlinearity. A number of simple and comprehensive models possess such nonlinearity and thus are able to reproduce the late Quaternary glacial cycles in good agreement with the paleodata. One of the main tasks of the GMT is to explain how such nonlinear responses originate from known processes and feedbacks operating in the Earth system, in particular, what determines the critical size of the Northern

Hemisphere ice sheets by reaching which they rapidly melt and disintegrate in response to rising boreal summer insolation. Understanding which processes in the Earth system explains such response is one of the main challenges of contemporary Quaternary climate dynamics. The mechanism of this instability, named a "domino effect" by analogy with a simple mechanical system, is discussed above and based on the results of CLIMBER-2 and several other studies. Due to the complexity of the processes associated with glacial terminations and the absence of their analogies in present-day climate, a

better understanding of the mechanism of glacial terminations requires further studies with comprehensive ESMs.

The processes which convert the seasonal and regional changes in insolation into global climate changes are now reasonably understood, and Arrhenius' idea about the role of $CO_2$ in ice ages is confirmed. Modelling studies show the lowering of $CO_2$ by ca 100 ppm reinforced by $CH_4$ and $N_2O$ drops explain about half of the maximum glacial cooling, currently estimated at 5-6ºC, while the rest is attributed primarily to ice sheet expansion and sea level drop. It has been shown that $CO_2$ also plays

an important role in amplifying glacial cycles. This aspect of glacial cycles is still lacking a satisfactory explanation. Most studies to date were devoted to simulations of the low $CO_2$ level at LGM, and the $CO_2$ radiative forcing has been prescribed in such studies. While such simulations are useful for testing the capability of modern ESMs to reproduce carbon cycle response to glacial climate forcings, they do not answer the question of how the ice sheet-climate-carbon cycle feedback loops operated during the late Quaternary. The "stew" of the processes playing a role in glacial-interglacial $CO_2$ variability has been

discussed but still remains tentative and requires further studies with the comprehensive earth system models.

Another widely discussed challenge for the GMT is the nature of the 41 kyr world and the mechanism of the MPT. The difference between the early Quaternary 41 kyr world and the late Quaternary 100 kyr worlds can be solely explained by a gradual increase in the critical size of the ice sheets. As a result, glacial cycles, which are initially locked to obliquity cycles, became locked to the shortest eccentricity cycle. The most straightforward explanation for the increase in critical size is the

removal of the terrestrial sediments from the northern continents by glacial erosion. This mechanism alone allows reproducing the relatively sharp transition from 41 kyr to 100 kyr world in the models. The only obvious discrepancy with reality is that in all model simulations, the 41 kyr world contains a clear precession presence while paleoclimate proxies usually show very little of such presence. The explanations for the lack of precession in the 41-kyr world range from physical mechanism (antiphase response to the precession of the ice sheets in the northern and southern hemispheres), interpretation problems

(much larger contribution to $\delta^{18}O$ temperature signal), or some problems with the processing of paleoclimate data (like orbital tuning). In any case, this is an interesting but not critical issue for understanding how the Earth system responds to the orbital forcing.



The comprehensive understanding of the mechanism of Quaternary climate variability still represents a formidable scientific challenge since this is a genuinely multidisciplinary problem. It is additionally complicated by the fact that present-day observational data provides insufficient constraints on the models used for paleoclimate studies, while paleoclimate records usually contain only indirect and incomplete information about past climate changes. Nevertheless, almost a century following the publication of Milankovitch's fundamental work, has a framework for the Generalized Milankovitch Theory emerged.

*Code and data availability.* The source codes and the output of the model simulations is available on request from the author.

*Competing interests.* The author declares that he has no conflict of interest.

*Acknowledgements.* The author is thankful to Matteo Willeit and Christine Kaufhold for helpful discussions.



**Appendix A**

**A.1 Different metrics for orbital forcing**

As has been shown for the first time by M. Milankovitch, changes in eccentricity, obliquity, and precession cause the substantial redistribution of insolation between latitudes and seasons. The comprehensive visualization of "orbital forcing" requires a 3-D plot (latitude - day of the year – time in years before/after present) which is impractical. Instead, different metrics for the orbital forcing were proposed. Milutin Milankovitch used as a metric for orbital forcing the so-called caloric summer half-year insolation, i.e. insolation integrated over the half-year with the highest insolation. In the post-Milankovitch time, it became common to use June or July insolation at 65°N (e.g., Berger 1978), although caloric half-year insolation is also used (e.g., Tzedakis et al., 2017). It is important to realize that using the modern calendar for defining orbital forcing on orbital time scales is problematic because the same calendar month at different years corresponds to different Earth's positions relative to the perihelion and aphelion (e.g., Kutzbach and Gallimore, 1988). This is why it is advisable to define the metric for orbital forcing in a way that does not depend on the choice of calendar. Examples for such definitions are maximum summer insolation used in this paper or insolation averaged over a certain period of time, e.g., caloric half-year insolation originally used by Milankovitch. Examples of different insolation curves for 65°N are shown in Fig. A1.

Experiments with the CLIMBER-2 model show that during the Late Quaternary, the rate of changes (dv/dt) of the Northern Hemisphere ice volume, which is primarily related to the surface mass balance of ice sheets, is highest correlated with the maximum summer insolation at 65°N. When performing the same analysis but for reconstructed global sea level (Spratt and Lisiecki, 2016), which represent the global ice volume, the best correlation is found for the 65°N insolation average over the four months with the highest insolation (i.e. roughly MJJA). However, as Fig. A1 shows, the difference between maximum summer insolation and 4-month insolation is relatively small in obliquity and precession relative contribution. It is thus important to note that neither our model nor data suggest that caloric summer half-year insolation is the best proxy for orbital forcing. This can be explained by the fact that most of the ice melt (both at present and during glacial cycles) occurs only during a few summer months. Both precession and obliquity play essential roles in all computed insolation curves. The quantitative difference is that for the caloric half-year insolation, the contributions of precession and obliquity are comparable most of the time. In contrast, for the maximum summer insolation, the obliquity component is comparable with precession only during periods of a relatively low eccentricity.

However, there is one metric of orbital forcing which contains very little precession variability and is often used for the interpretation of paleoclimate records. This is the so-called "integrated summer insolation" proposed by Huybers (2006) and which is defined as

$$S = \sum_i I_i \text{ for } I_i > I_0,$$

where $I_i$ is the daily insolation at 65°N and $I_0$ is a constant value set to 275 W/m$^2$ in Huybers (2006). This choice of the metric for orbital forcing is justified by the fact that seasonal variations of temperature over land and insolation are rather similar





(with about one month lag of temperature behind insolation) and that the melt rate of ice sheets is often parameterized through the so-called Positive Degree Days index defined as $PDD = \sum_i T_i$ for all daily temperatures $T_i > 0$ (Fig. A2) . However, to correctly translate insolation into PDD index, one should consider the fact that for an arbitrary temperature scale, PDD should

be defined as $PDD = \sum_i (T_i - T_0)$ . Only for the Celsius temperature scale $T_0 = 0°C$ and, thus $T_0$ can be omitted in the formula for PDD. However, for example, in the case of Kelvin temperature scale $T_0 = 273.15K$. Therefore, PDD should be expressed through the insolation as

$$S = \sum_i (I_i - I_0) \text{ for } I_i > I_0 \quad . \tag{A1}$$

Obviously, this formulation is significantly different from the original "integrated summer insolation" (compare figures A1

(d) and (e)) and similarly to other metrics for orbital forcing it contains a strong precession signal. Thus, none of the physically meaningful metrics for orbital forcing is free of a strong precession component which is important for discussions of the nature of 41 kyr world.

### A2. Imbrie and Imbrie conceptual model

One of the first and well-known conceptual models of late Quaternary glacial cycles is the Imbrie and Imbrie (1980) model.

This model has been applied for the "orbital tuning" of the widely used LR04 record (Lisiecki and Raymo, 2005). The model is represented by a single equation for global ice volume $v$ (in arbitrary units):

$$\frac{dv}{dt} = \frac{V_e - v}{\tau_k},$$

where $V_e = 1 - c_1 f$ is the equilibrium volume for orbital forcing $f$, and $c_1$ are constants. The relaxation time scale $\tau_k$ depends on the regime: $k=1$ if $v > V_e$ and, $k=2$ if $v < V_e$. If $\tau_1 = \tau_2$, the model is linear and its response to orbital forcing is similar to

forcing itself with only obliquity and precession frequencies present. Using $\tau_2$ much larger than $\tau_1$, as in Imbrie and Imbrie (1980), makes the model nonlinear. The model performance is far from perfect (see, e.g. Paillard, 2001), and even for the optimal choice of model parameters, the correlation between modelled $v$ and LR04 stack for the last 800 kyr does not exceed 0.6. Even more important is that the modelled ice volume variability has too much energy in 400 kyr band and too little energy in 100 kyr band (Fig. A3).

### A3. The rules for the timing of glacial terminations based on 2 or 3 obliquity cycles and "effective energy"

Huybers and Wunsch (2005) analyzed benthic $\delta^{18}O$ records and concluded that glacial terminations of the late Quaternary spaced in time by two or three obliquity cycles while the link with precession and eccentricity were not found to be statistically significant. Admittedly, later Huybers (2009) admitted the role of precession in glacial cycles, but the concept of "two or three



obliquity cycles" remains rather popular. Indeed, there is a clear tendency for glacial terminations to occur during periods of
positive obliquity anomalies, although the durations between individual terminations can deviate significantly from two (82)
or three (123 kyr) obliquity cycles (see below). When rounding to the nearest integers, the last eight glacial cycles can be
expressed in terms of obliquity periods as 2, 2, 2, 3, 2, 2, 3, 3. The average number of obliquity periods 2.6 multiplied by
obliquity periods gives 97 kyr, which is very close to the observed dominant periodicity of late Quaternary glacial cycles. But
it is also clear that the probability of getting the correct sequence of "2" and "3" by random choice is 1/256 =0.004, which is
a rather low probability, while the models used in this study demonstrate that the timing of all glacial terminations of the late
Quaternary is relatively robust. Even more problematic for the "2 or 3" hypothesis is that if this hypothesis is correct, then the
duration of individual glacial cycles should cluster around 82 and 123 kyr with the equal probabilities of being longer or shorter
2(3)*41 kyr. In fact, durations of the last eight glacial cycles are rather uniformly distributed between ca 80 and 120 ka, with
five of eight glacial cycles having durations between 90 and 110 kyr, i.e. closer to 100 kyr than to 2 or 3 obliquity cycles (e.g.
Konijnendijk et al. 2015).

The term "effective energy" was introduced by Tzedakis et al. (2017). According to this paper, terminations (during the
last 600 kyr) occur when the "effective energy" defined as $I_{peak}+b(D-t)$ exceeds 6.412 GJ/m². Here $I_{peak}$ is the magnitude of the
maximum caloric half-year summer insolation at 65ºN, $D-t$ is the time since previous deglaciation and $b$ is a constant. This
rule for glacial terminations implies that the Earth system somehow "knows" in advance whether the effective energy will
cross the threshold value, which usually happens well after the beginning of terminations. In some cases (for example, for the
penultimate glaciation), glacial terminations have been completed even before the maximum of the "effective energy" has
been reached. How this rule and a somewhat similar one described in Huybers (2012) can be reconciled with the GMT, which
"predicts" terminations not based on the magnitude of insolation maxima, but rather by the timing when ice volume exceeds
the critical threshold?   In fact, it can be shown that, although these rules are based on completely different physical concepts,
they indeed give similar timing of glacial terminations. The rule based on the "efficient energy" formulated by Tzedakis et al.
(2017) implies that glacial terminations occur only during those maxima of caloric half-year summer insolation which are
above the average value of insolation maxima which occurred close to each obliquity maxima. This can only happen if both
precession and obliquity components of summer insolation are positive, and the eccentricity is also above average. Since the
period of obliquity (41 kyr) is approximately twice longer rhan the period of precession and is comparable with a half of 100
kyr eccentricity period, it is highly probable that the previous positive precessional maximum occurred during the obliquity
minimum and eccentricity was below average. But this is precisely the condition for reaching the critical ice volume in GMT.
Thus, with the appropriate initial conditions, Tzedakis et al. (2017) rule produces the same (or similar) sequence of terminations
as the GMT.

**A4. Model 2**

Model 2 is a one-equation conceptual model aimed at reproducing the principal CLIMBER-2 results. It is derived using
several assumptions, among which is that total accumulation (mass gain of ice sheets) $G$ is proportional to the ice sheet area,





but since the later closely followed global ice volume $v$ (e.g. Ganopolski et al., 2010), the total accumulation is parameterized as

$$G = a_1 v .$$ (A2)

The total mass loss (including both surface and basal melt, and ice calving into the ocean) $L$ is a nonlinear function of ice volume. It also includes an additional term proportional to the rate of ice volume change:

$$L = a_2 v + b v^2 - \delta g v^* \frac{dv}{dt} ,$$ (A3)

     The last term is nonzero only for ice sheets decay, i.e. $\delta = 1$ if $\dfrac{dv}{dt} \leq 0$ and $\delta = 0$ if $\dfrac{dv}{dt} > 0$. The value

$v^*(t) = \dfrac{1}{N} \displaystyle\int_{t-N}^{t} v(\tau)\, d\tau$ is the memory term defined as the average ice volume over the previous N kiloyears. This term is crucial

to simulate glacial terminations through the "domino effect. The effect of insolation M (which can be positive and negative) is accounted for through a linear function of orbital forcing $f$ (anomaly of maximum summer insolation at 65ºN, see Section 4):

$$M = c f' ,$$ (A4)

     where $f' = f + f_1$ , and $f_1$ is the model parameter. Of course, orbital forcing can contribute both to "mass loss and gain and

is treated here separately just for convenience.

     Temporal evolution of global ice volume (in normalized units) is thus described by the equation

$$\frac{dv}{dt} = G - L + M = \left( av - bv^2 - cf' \right) D^{-1} ,$$ (A5)

     Where $a = a_1 - a_2$, and $D = 1 - \delta g v^*$ .

     To avoid the denominator in Eq. A4 from approaching zero and subsequently becomingnegative, it is additionally required

that $D \geq \varepsilon$. In these equations, $a$, $b$, $c$, $g$, $\varepsilon$ and $N$ are model parameters (see Table 1). Mathematically this equation is not much more complex than governing equation of the Model 3, but it has a clearer physical meaning. At the same time, it is easy to show that Model 3 is a simplified version of the Model 2. The equilibrium solution of eq (A5) for ice volume $V_e$ (see also Fig. A4):

$$V_e = \left( a \pm \left( a^2 - 4bcf' \right)^{1/2} \right)(2b)^{-1} .$$ (A6)

This is a rotated counterclockwise parabolic curve, identical to the equilibrium solutions of Model 3 (Fig. 4).

     As it follows directly from Eq. (A4), growth of ice volume from zero (glacial inception) can only occur for $f' < 0$, i.e. $f < -f_1$ , which means that the value $f_1$ (bifurcation point B₁) has the same meaning as $f_1$ in Model 3. The position of the second

bifurcation point B₂ in the phase space of orbital forcing (the equivalent of $f_2$ in Model 3) is defined as $f_2 = \dfrac{a^2}{4bc} - f_1$ . Since





the vertical scale of equilibrium solution is defined by the combination of parameters $a$ and $b$, the number of parameters can

be reduced to six by setting $b=a/2$. This gives the equilibrium solution $V_e(f_2)=1$ for ice volume at the bifurcation point $B_2$, which is the same as in Model 3. While Model 2 has qualitatively the same stability diagram as Model 3, it differs from the latter in that the time scale of relaxation towards the equilibrium solutions in Model 2 is not constant and depends on the position in the insolation-volume phase space. A typical relaxation time scale of Moldel 2 in the glaciation regime (i.e. when $dv/dt > 0$) can be estimated as $2a^{-1}$ which, for the value of $a$ given in Table 1, is ca 27 kyr, i.e. is similar to the (constant)

relaxation time scale of Model 3. In the case of glacial termination, the rate of ice loss increases by the factor $D^{-1}$, which is an order of ten when the memory term $v^*$ approaches the critical ice volume equal to $g^{-1}$.

Fig. A2 shows results of simulations of the last 800 kyr in comparison with paleoclimate reconstructions for the optimal set of model parameters (see Table 2). Correlation between model and LR04 stack is 0.8 which is similar to Model 3. As in the case of Model 3, the "termination" regime described by the term $D$ in Eq. A4 is absolutely crucial. Without this term,

Model 2 simulates a weakly nonlinear response with the presence of all Milankovitch frequencies, similar to the one-regime version of Model 3 and the Imbrie and Imbrie model. Compared to Model 3, Model 2 does a better job for the pre-MBT glacial cycles but, similarly to CLIMBER-2 (Willeit et al., 2019), has a problem to simulate long interglacial MIS11 which the Model 3 does not. This is explained by a very weak orbital forcing which triggered Termination 5 (transition to MIS 11 interglacial). Since in Model 3, the transition to glaciation regime does not depend on the value of orbital forcing (it only should be growing

and positive), it is easier to simulate the correct timing of Termination 5 with Model 3 than with Model 2, where the magnitude of orbital forcing matters. After MIS 11, Model 2 performs well for a rather broad range of model parameters and the maximum correlation between simulated ice volume and LR04 for this time interval reaches 0.9. Note that the extended version of Model 2, which also includes equations for $CO_2$ and global temperature, has been developed and applied to future glacial cycles simulation in Talento and Ganopolski (2021).

## A5. The role of obliquity in 100 kyr world

Since obliquity completely dominates $\delta^{18}O$ records of the 41-kyr world, it is natural to assume that obliquity also plays an essential role in 100 kyr world. However, the experiments performed with the CLIMBER-2 model (Ganopolski and Calov, 2011) demonstrate that the dominant 100 kyr periodicity is present even when the obliquity component is completely excluded from the orbital forcing. At the same time, this periodicity cannot be simulated without eccentricity modulation of the

precession component. Model 3 demonstrates the same behaviour: when forced by the precession component of orbital forcing alone, it simulates a strong maximum in the frequency spectrum at 100 kyr for a broad range of the model parameter $V_c$. The presence of obliquity affects the timing of some glacial terminations, and therefore the duration of the individual glacial cycles, but not the dominant periodicity (Fig. A6). In the absence of the obliquity component, the durations of individual glacial cycles cluster around four and five precession periods (i.e., ca. 90 and 100 kyr) rather than two and three obliquity periods. Similar

to CLIMBER-2 (Ganopolski and Calov, 2011), in Model 3 with obliquity-free orbital forcing, more energy in frequency spectra





is seen in the 400 kyr band than in the case of realistic orbital forcing. However, the amount of energy in the 400-kyr band, unlike the 100 kyr, is strongly dependent on the choice of model parameters. In short, model simulations show that, although the obliquity component of orbital forcing plays an important role in climate variability booth during the 41-kyr and 100 kyr worlds, it has nothing to do with the dominant 100 kyr periodicity of the glacial cycles of the late Quaternary.

**A6. Orbital tuning and the absence of precession in early Quaternary records**

One of the potential causes for the absence of precession in the $\delta^{18}O$ spectra during early Quaternary is that these time series are often orbitally tuned using obliquity as the target (e.g. Lisiecki and Raymo 2005). Since obliquity is the dominant frequency in the Early Quaternary ice volume variations, such a choice seems to be reasonable. However, in the case of a strongly nonlinear system, tuning the age model to maximize one frequency may lead to the suppression of another one. This is

illustrated by Fig. A7, where the results of model simulation described in Section 4 and shown in Fig. 11b have been "tuned to obliquity". Simulated ice volume already has a strong obliquity component (Fig. A4c) and the correlation between simulated volume and (negative) obliquity anomaly shifted by 5000 years is relatively about 0.5. However, the spectrum has also pronounced precession maxima. To mimic potential impact of orbital tuning on the frequency spectrum, the "age" of modelled ice volume has been modified to maximize correlation with obliquity. This time correction satisfies two criteria similar to that

used in the actual orbital tuning: (i) the initial (in this case, real) "age" has not been changed by more than 10 kyr, and (ii) the original time scale was not stretched or squeezed by more than a factor two at any time. Fig A4c shows the change of the original "age," which gives the higher correlation between "tuned" volume and the target (obliquity). Now, this correlation reaches 0.65, practically the same value as the correlation between LR04 stack and obliquity (0.66). As Fig A7 show, this "tuning" slightly increases spectral energy in the obliquity band but causes a nearly complete disappearance of precession

components (Fig. A4e).



**Table 1**. Parameters of conceptual models used in the paper

| Model | Optimal parameters |
|---|---|
| **IIM80** | $t_1$=24 kyr , $t_2$=3 kyr, $c_1$=0.02 W$^{-1}$ m$^2$ kyr$^{-1}$ |
| **MiM** | $t_1$=65.6 kyr, $t_2$=35.6 kyr, $c_1$=0.00126 W$^{-1}$ m$^2$ kyr$^{-1}$ |
| **Model 2** | $a$=0.075 kyr$^{-1}$, $c$=0.0018 W$^{-1}$ m$^2$ kyr$^{-1}$, $g$=0.6, $\varepsilon$=0.03, $f_1$=3.6 W m$^{-2}$, $N$=25 kyr |
| **Model 3** | $t_1$=30 kyr, $t_2$=10 kyr, $f_1$= -16 W/m$^2$, $f_2$=16 W/m$^2$, $v_c$=1.4 |





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



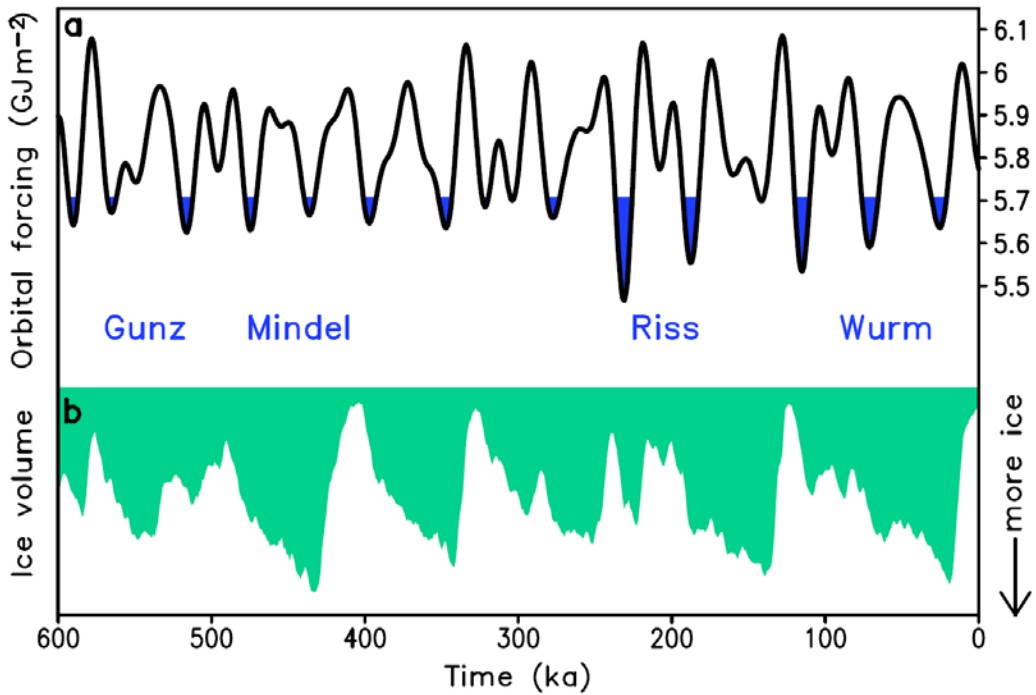

**Figure 1.** Milankovitch forcing and ice volume variations over the past 600 kyr. (a) Caloric summer half-year insolation at 65°N. Blue shading represents insolation below 5.7 GJ m$^{-2}$, the insolation threshold selected to match best Fig. 57 in Milankovitch (1941). According to

the Milankovitch theory, glaciations occurred during periods of low insolation. The names of major glaciations, according to Penck and Brückner (1909), are written below the "Milankovitch curve". (b) LR04 (Lisiecki and Raymo, 2005) benthic $\delta^{18}O$ stack, a proxy for the global ice volume (inversed for convenience).

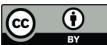



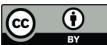

**Figure 2**. Orbital forcing and the Earth system response during the entire Quaternary period. (a) Maximum summer insolation at 65°N computed using Laskar et al. (2004) orbital parameters; (b) LR04 (Lisiecki and Raymo, 2005) benthic $\delta^{18}O$ stack, a proxy for global ice volume; (c) wavelet spectra of LR04 stack (in arbitrary units).



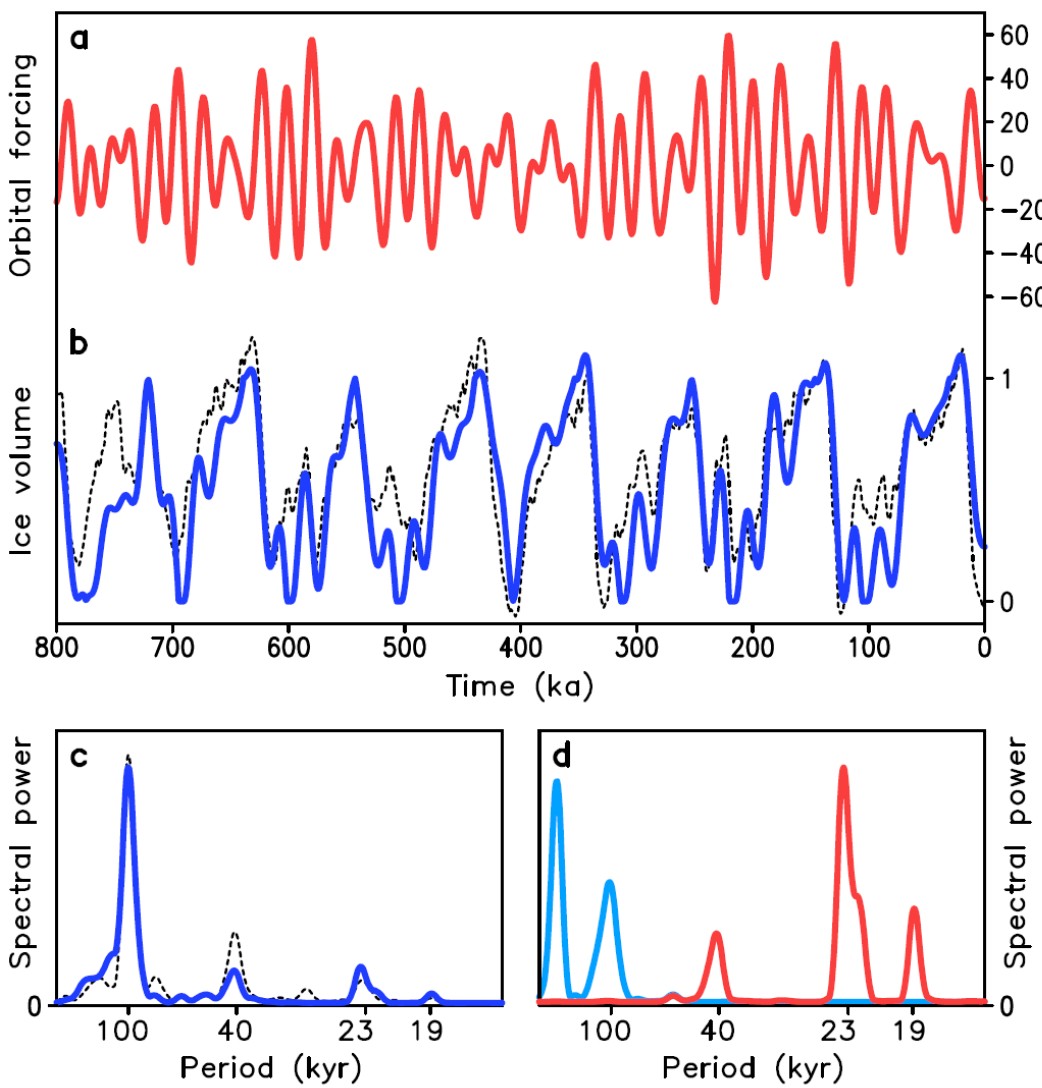

**Figure 3.** Simulations of late Quaternary glacial cycles with the "minimal" conceptual models MiM. (a) orbital forcing (maximum summer insolation at 65ºN), (b) results of MiM versus paleoclimate reconstruction. The solid lines are modelling results, and the dashed line is arbitrary scaled LR04 δ18O stack. (c) frequency spectra of MiM in comparison with the frequency spectra of LR04 stack. (d) frequency spectra of orbital forcing (red) and eccentricity (blue).

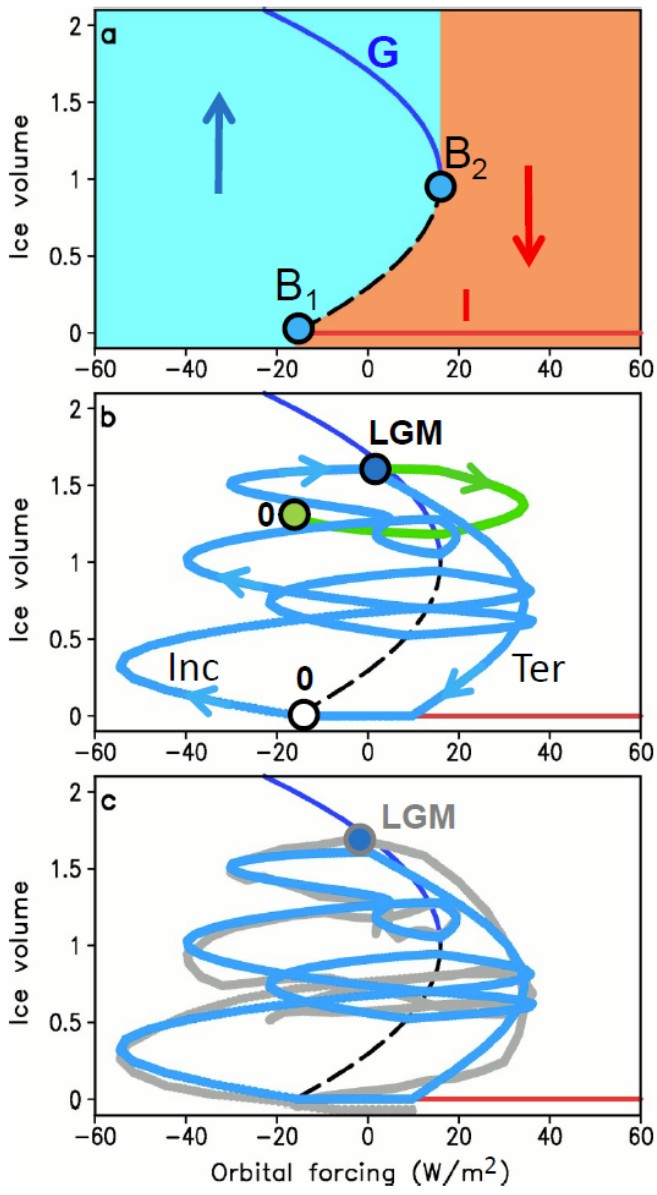

**Figure 4** Evolution of the modelled ice volume in the phase space of orbital forcing. a) Two equilibrium solutions: G – glacial, I –interglacial. The blue area is the attraction domain of the glacial state, the red area is the attraction domain of the interglacial state, dashed line is an unstable solution separating the two domains. $B_1$ and $B_2$ are the bifurcation points, $B_1$ corresponds to glacial inception. This diagram corresponds only to the glaciation regime. For the deglaciation regime, there is only one equilibrium state – I. b) Evolution of simulated ice volume (nondimensional) during the past glacial cycle starting at time 120 ka. "Inc" denotes glacial inception, "Ter" - glacial terminations. Note that after the LGM (20 ka) the model without termination regime (green line) failed to deglaciate. (c) Comparison of model simulation with LR04 benthic $\delta^{18}O$ stack.



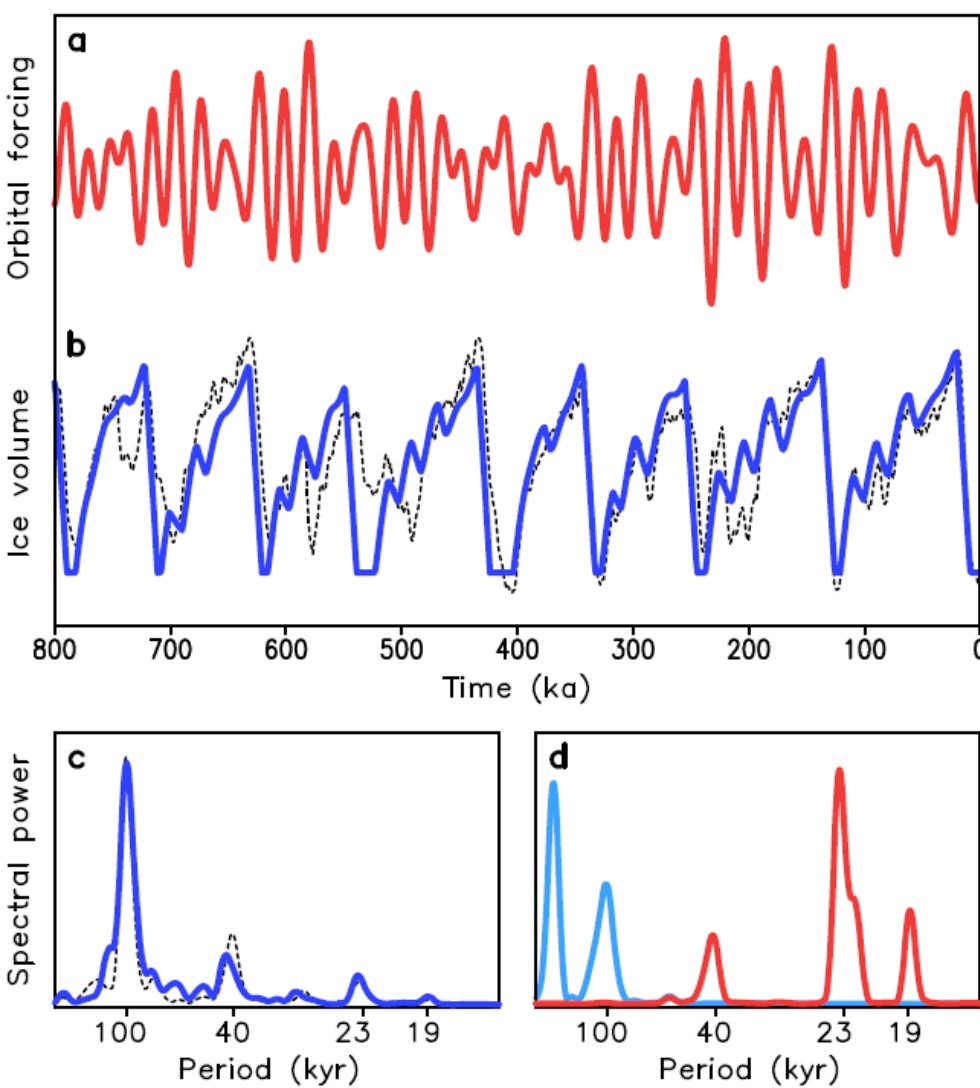

**Figure 5** Results of simulations of the last 800 kyr with the conceptual Model 3. (a) Orbital forcing; (b) Simulated ice volume in nondimensional units (blue) and scaled LR04 stack (dashed line); (c) Spectra of simulated ice volume (blue) and LR04 stack (dashed line); (d) frequency spectra of the orbital forcing.



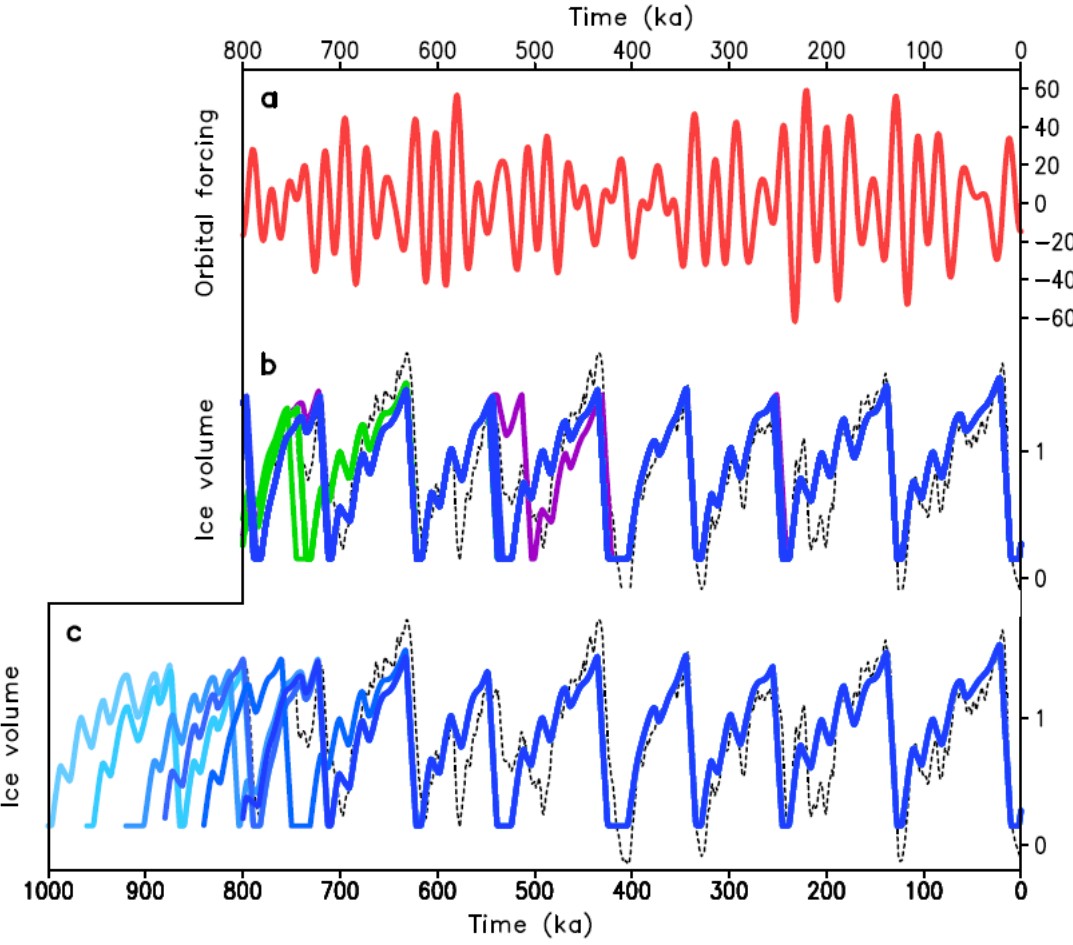

**Figure 6.** Results of simulations of the last 800 (1000) kyr with the conceptual Model 3. (a) Orbital forcing; (b) Simulated ice volume (green lines correspond to the range of $V_c$ 1.32-1.38, blue to the range 1.39-1.48, purple to $V_c$=1.49). Thus, the timing of most glacial terminations (and all the latest five) coincide for the broad range of $V_c$ from 1.32 to 1.49. (c) Simulated ice volume for $V_c$=1.4 (different colours correspond to different starting times from 1000 to 800 ka with the step 40 ka). All runs converge to the same solution after less than 200 kyr. Note the similarities with Fig. 6b from Ganopolski and Calov (2011). (b, c) LR04 stack is shown by dashed lines.







**Figure 7.** Simulation of glacial cycles with the Model 3 for different artifitial orbital forcings. (a, b) forcings, (c, d) simulated volume in arbitrary units. (e, f) frequency spectra of simulated ice volume. (a, c, e) Periodic forcing with A=25 W/m$^2$ , T=23 kyr and $\varepsilon$=0. (Eq. 3). (b, d, f) Periodic forcing with additional 100 kyr amplitude modulation ($\varepsilon$ = 0.5). Green lines correspond to $v_c$=1.2, blue – $v_c$=1.33, purple – $v_c$=1.47.





**Figure 8.** Simulation of the last 2.8 Ma with Model 3 ($V_c$=1.3). a) Eccentricity, b) simulated ice volume (arbitrary units), (c) frequency spectrum of eccentricity, (d) spectrum of ice volume simulated with Model 3 (blue) and the Imbrie and Imbrie model (magneta), (e) histrogram of the durations of individual glacial cycles simulated with Model 3.



**Figure 9.** Simulation of the last 2800 ka with constant critical ice volume parameters ($V_c$=1.3). a) Eccentricity, b) simulated ice volume (arbitrary units), c) wavelet spectra of eccentricity, and d) wavelet spectra of simulated ice volume.

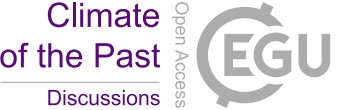

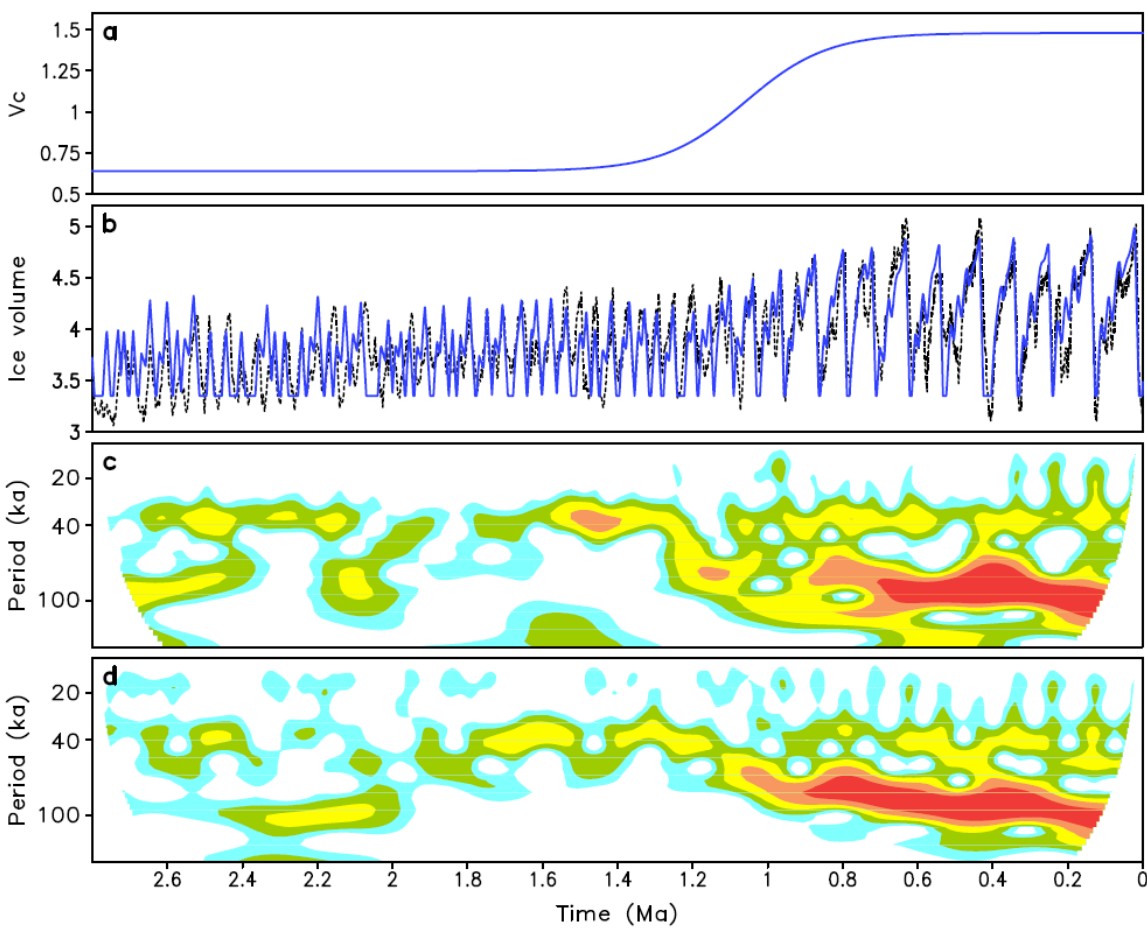

**Figure 10.** Simulations of MPT with the Model 3. a) Temporal dependence of critical ice volume $v_c$; b) Simulated ice volume (blue) versus LR04 (dashed); c) Wavelet spectra of LR04 stack; d) wavelet spectra of modelled ice volume.





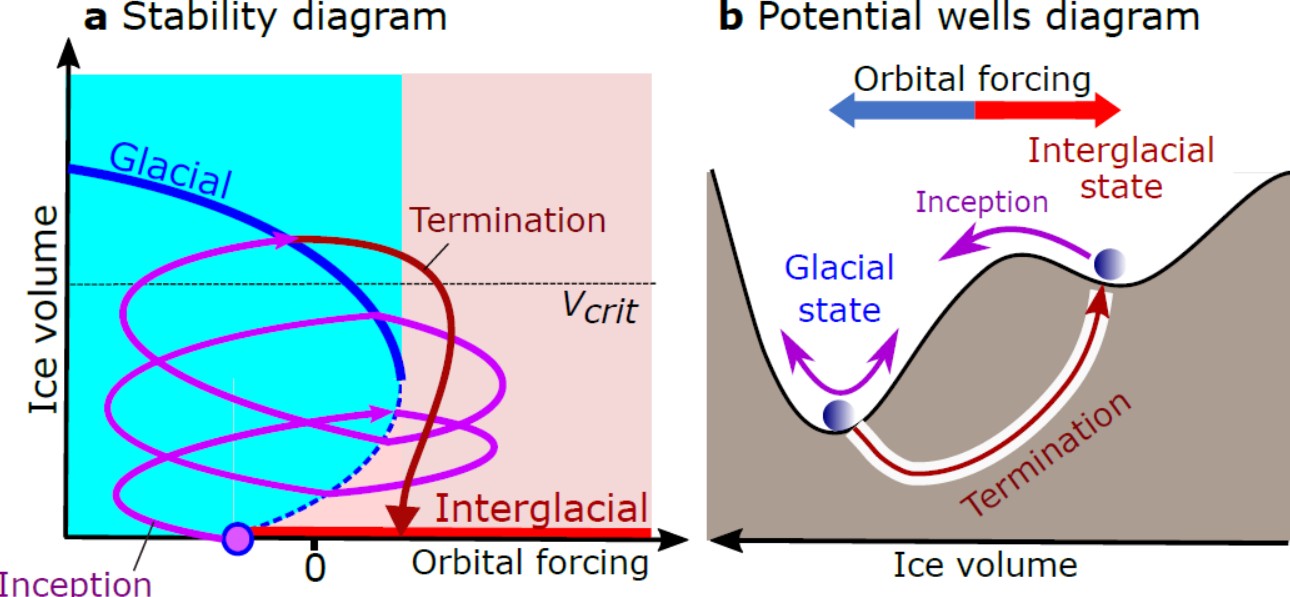

**Figure 11**. Generalized stability diagram of the climate-cryosphere system. a) Phase portrait in orbital forsing (anomaly of maximum summer
insolation at 65°N) space. The notations are the same as in Fig. 4a. b) two potential wells represent the glacial and interglacial states. The
diagram corresponds to the average (zero) orbital forcing, while orbital forcing is considered here as the external perturbation which moves
the system from one equilibrium to another one. Glacial termination is depicted as the tunnel transition under the potential barrier separating
two stable states.



1565

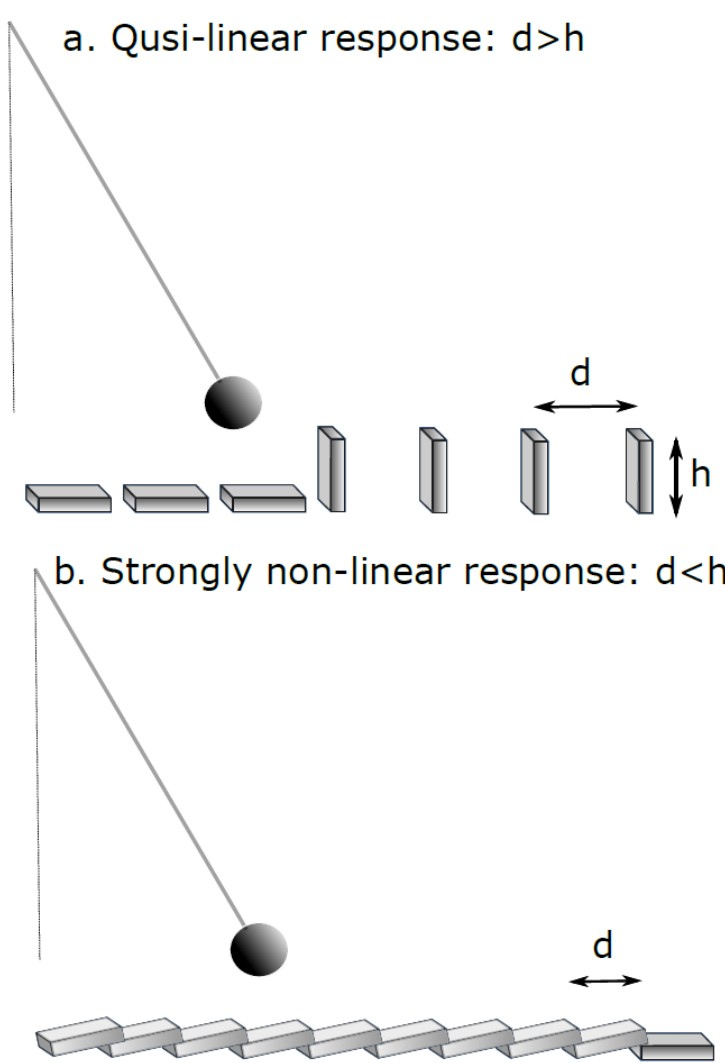

**Figure 12.** The domino effect.



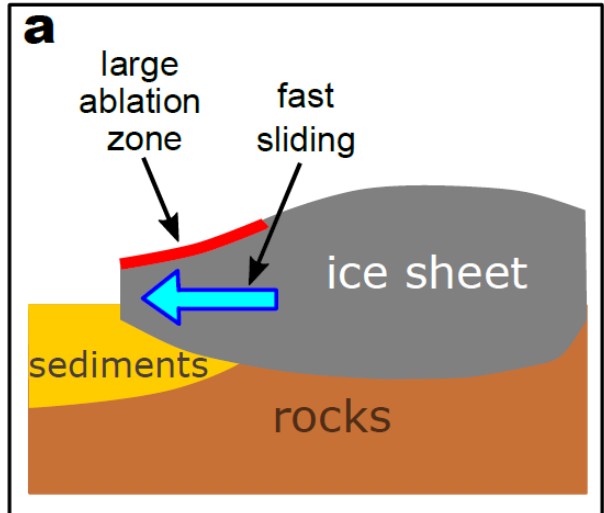

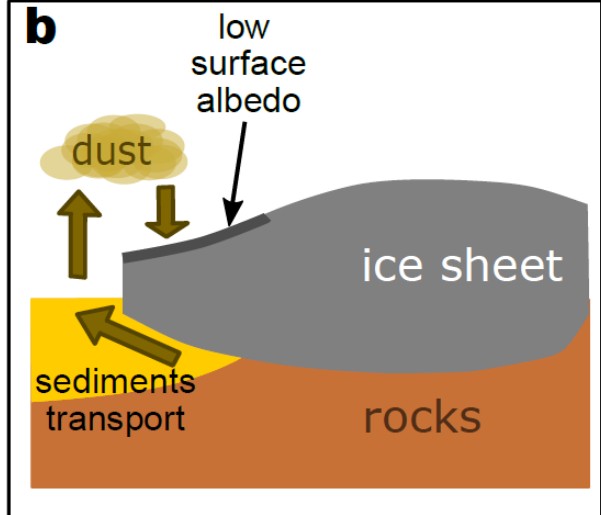

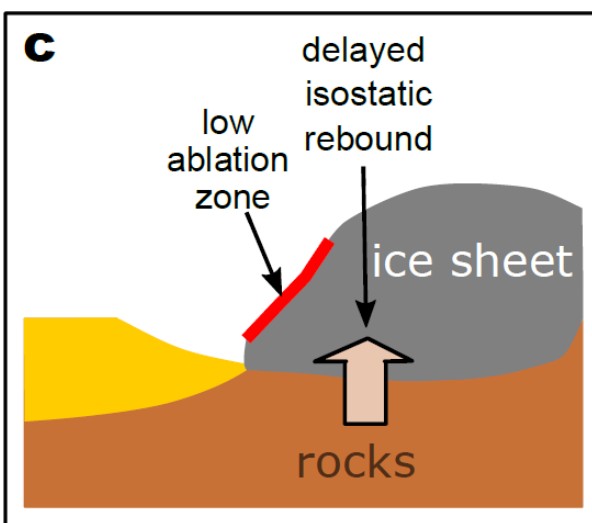

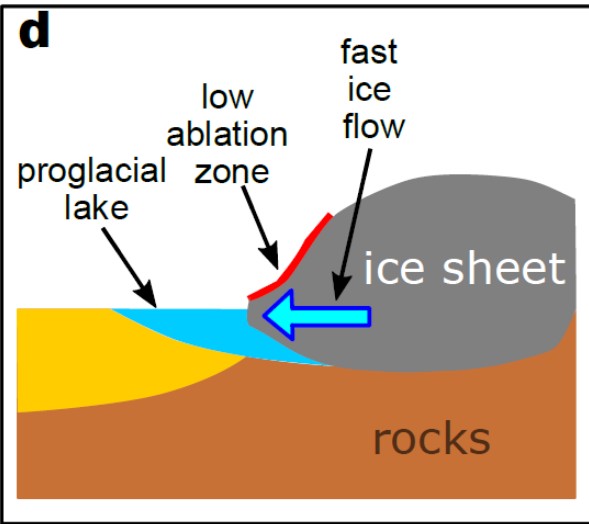

**Figure 13.** The principal elements of the termination mechanism: a) faster ice sliding and lower elevation over sediment area; b) effect of glaciogenic dust deposition on snow albedo and surface mass balance; c) effect of delayed bedrock relaxation on the altitude of ablation zone; d) effect of proglacial lakes on ice sheet mass loss.





**Figure 14.** The process-based conceptual model of GMT. Blue circles represent components of the climate system and green – global carbon cycle. This flow chart diagram includes only processes and components of the Earth system which were explicitly treated in our simulations of glacial cycles with the CLIMBER-2 model and which we found sufficiently important for accurate simulations of glacial cycles. Ocean circulation and ventilation are combined in one process. To improve readability, the diagram is simplified and some feedbacks are missed. Abbreviations: *SAT* –surface air temperature, *SMB* – surface mass balance of ice sheets, *ALK* –alkalinity. Ocean circulation and ventilation are combined also includes ocean ventilation.





**Figure 15.** A simple rule to determine the timing of glacial terminations. Grey shaded areas correspond to periods of low eccentricity, light blue to the periods of low obliquity during low eccentricity, red shading on the precession forcing curve highlights positive precession anomalies satisfying GMT criteria, and blue shading corresponds to following precession minima during which critical ice volume reached (purple vertical line) and deglaciations start soon after these insolation minima. Orange shading on the precession curve corresponds to "false" positives, i.e. the cases when conditions of the Termination Rule are met, but they occurred earlier than 60 kyr after the previous termination. The lower curve is LR04 benthic $\delta^{18}$O stack.

**Figure A1**. Different proxies for orbital forcing. (a-e) insolation at 65ºN during the past 800 kyr and (f-j) corresponding frequency spectra. (a, f) Maximum summer insolation, (b, g) insolation averaged over four months with the highest insolation; (c, h) insolation averaged over 6 months with the highest insolation (the equivalent of caloric half years energy but in different units), (d, j) "summer energy" computed according to Huybers (2006); (e, j) "summer energy" computed according to Eq. (A1). Insolation (a-c) are in W/m$^2$, "summer energy" (d) and (e) are in KJ. Frequency spectra are in arbitrary units.

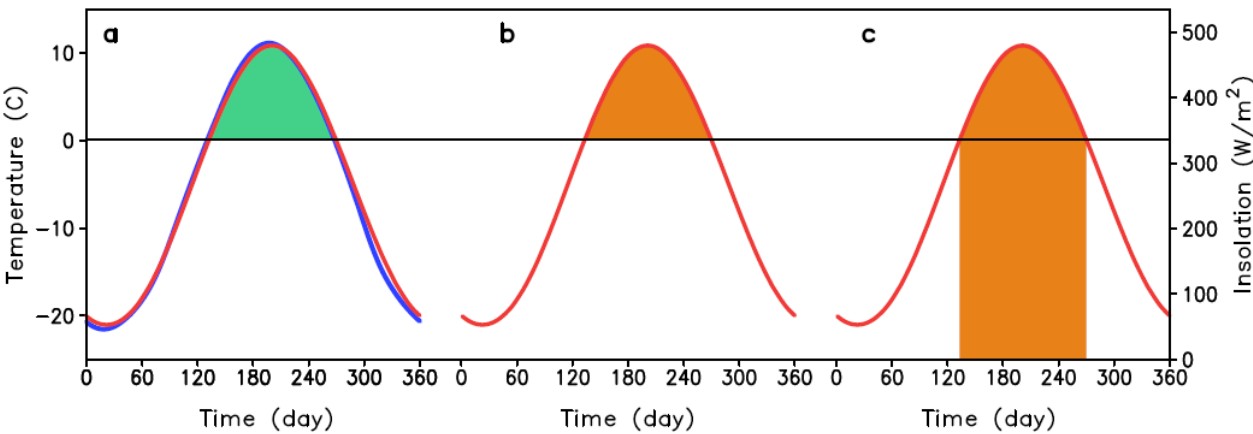

**Figure A2.** Present temperature and insolation curves at 65°N and different definitions of "summer energy". (a) Seasonal surface air temperature variation over land (blue) and insolation shifted by 30 days (red). The green area below the temperature curves represents the Positive Degree Day index. (b) The Orange area here represents an analogy for the positive degree days. The evolution of this characteristic is shown in Fig. A1e. This is a correct definition of summer energy. (c) The definition of "summer energy" according to Huybers (2006) is shown in Fig A1d. The horizontal axis is the time of the year in days.





**Figure. A3** Results of simulations of the last 800 kyr with the Imbrie and Imbrie (1982) model. (a) Orbital forcing; (b) simulated ice volume (solid) and LR04 stack (dashed); (c) frequency spectra of simulated ice volume (blue) and LR04 stack (dashed line); (d) frequency spectra of eccentricity (blue) and orbital forcing (red). Vertical scales are arbitrary.



**Figure A4**. (a) Stability diagram for Model 2. Notations are the same as in Fig. 3. (b) Trajectory of the system described by the Model 2 in the phase space of orbital forcing-ice volume for the last 125 kyr.



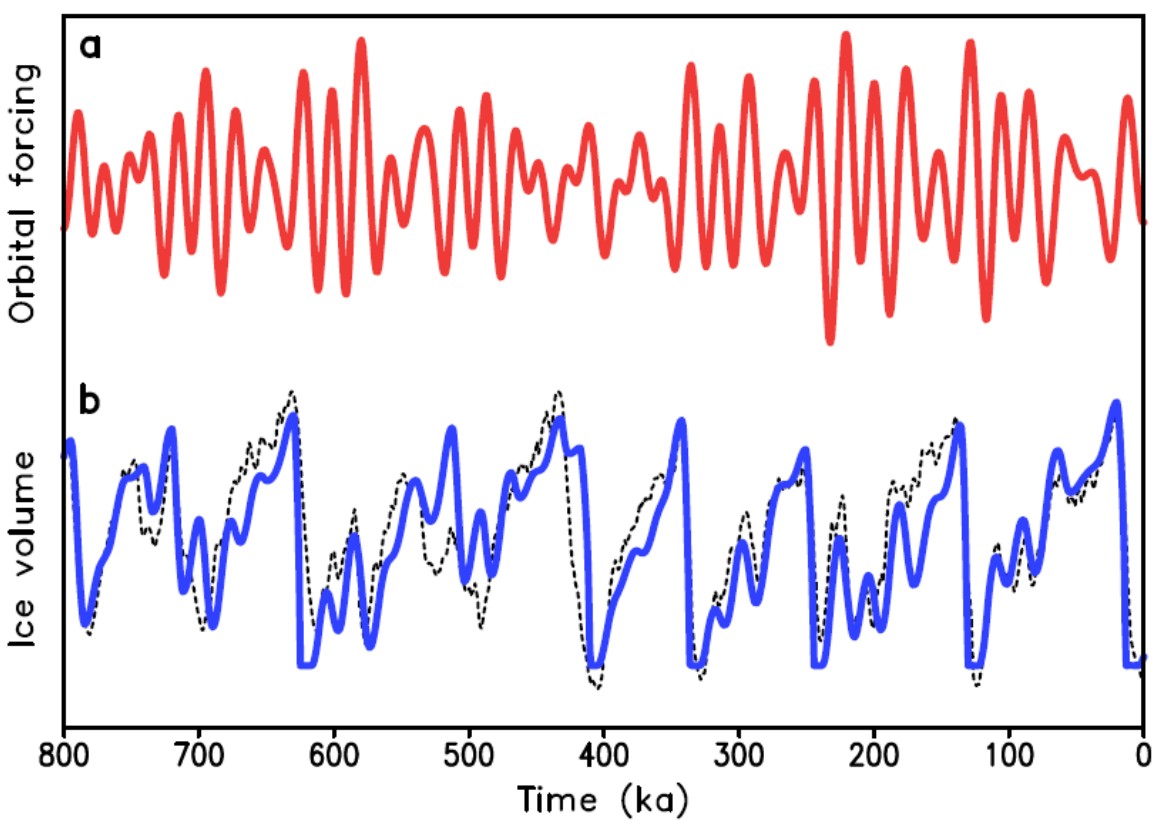

**Figure A5**. Simulation of late Quaternary glacial cycles with the Model 2.

**Figure A6.** Results of simulations of the last 800 kyr with the Model 3 with and without obliquity component in orbital forcing. (a) Full orbital forcing; (b) Orbital forcing without obliquity component; (c) Simulated ice volume with obliquity (blue) and without obliquity component (green), and LR04 stack (dashed); (d) Spectra of simulated ice volume with obliquity component (blue) and LR04 stack (dashed line); (e) Spectra of simulated ice volume without obliquity component (green) and LR04 stack (dashed line).

**Climate**
**of the Past**
Discussions
EGU

**Figure A7.** "Minimal orbital tuning". a) Tuning target: negative anomaly of obliquity shifted by 5000 kyr; b) simulated and "tuned" ice volume; c) time shift for the "tuned" time series; (d) and (e) corresponding frequency spectra. The dark blue line is the original simulation of the 41 kyr world with Model 3 (the same as in Fig 8) and the light blue line shows optimally "tuned" Model 3 simulations. The vertical scales in (d) and (e) are the same.