# Peer review of "Toward Generalized Milankovitch Theory (GMT)"

_Climate of the Past, 2023_

## Author Comment (AC1)

**Response to comments by Michel Crucifix (Referee #1)**

I thank the Referee for the useful and constructive comments and suggestions. Below is my response with the original Referee's comments in blue italic.

**Genera comments**

*It is relevant to consider the science reviewed in the present contribution as a development of Milankovitch's effort, because Milankovitch's purpose was to provide a theory of ice ages. Yet, I would argue, this is not the only possible way of 'generalizing' Milankovitch's theory. For example, a stream of modern research is focused on understanding orbital control of climate for other periods in the past, down to the Paleozoic. This other form of generalisation would give more importance to other climatic*

*and environmental effects of the orbital forcing than their effects on ice sheets, such as ocean circulation and nutrient supply. Therefore, I would suggest to be a bit more explicit about the Quaternary ice ages in the title (see suggestion below).*

As will be explained below, the Milankovitch theory has nothing to do with the climates of the Phanerozoic or any other period. It will be also explained why I do not like the term "*Quaternary ice*

*ages*".

*The word 'theory' is used in the present contribution to designate the ensemble of theoretical and empirical considerations that leads to our current understanding of the astronomical control on ice ages. The "theory" is illustrated by the "Model 3" conceptual model. The latter has obvious similarities with previously published models (especially from the Paillard's school) but the author explains that it is*

*formulated such as to be compatible with CLIMBER-2 output, along with general considerations about the bi-stability of ice sheets at the global level. The key point, here, is that the "Model 3" does not explicit introduces a 80ka time scale as Paillard 99 did, which in effect, puts more responsibility on the succession of precession cycles, modulated on eccentricity, to generate the 100ka signal. This definitely is an interesting point, and in that sense I can see why the author presents this work as a 'generalisation' of*

*Milankovitch's theory. Model 3 features a dynamics that wasn't in the original Milankovitch proposal.*

(I suppose the Referee means here and hereafter Paillard 98, not *Paillard 99*). I've got the impression that the Referee assumes that Model 3 is the GMT. It is not. The GMT is described in section 5: "Elements of GMT". To prevent such confusion, I will remove "GMT" from the title of Section 4, which describes Model 3.

*Yet, let us admit, many authors of models and theories over the last 3 decades have introduced a dynamical component, and in that sense generalize Milankovitch's theory.*

I acknowledged the important contributions of many authors toward the development of a comprehensive theory of glacial cycles. The paper refers to more than 100 publications of other authors. I do not claim that I am the first to try to "generalize" Milankovitch theory, i.e. to advance it beyond the classical Milankovitch theory to explain facts which were not known during the Milankovitch era. But I also believe that I advanced in this direction further than my fellow colleagues. Moreover, I have a "copyright" on the term GMT. I am not sure whether the Reviewer is aware of this fact, but I introduced this term in my Milankovitch Medal Lecture at EGU 12 years ago:

Geophysical Research Abstracts
Vol. 13, EGU2011-14170, 2011
EGU General Assembly 2011
© Author(s) 2011

[Figure]

**Towards Generalized Milankovitch Theory (Milutin Milankovic Medal Lecture)**

Andrey Ganopolski
Potsdam Institute for Climate Impact Research, Potsdam, Germany

In this lecture, the abstract of which is readily available, I already formulated a number of key elements of the GMT. In particular:

*(i) the glacial cycles represent a direct, strongly nonlinear response of the climate-cryosphere system to the orbital forcing; (ii) the strong 100 kyr cyclicity of ice volume variations originates from phase locking of the "long" glacial cycles to the eccentricity variations with 100 kyr periodicity, etc...*

I delayed the publication of this paper for such a long time because I wanted first to test my ideas with the results of the CLIMBER-2 model. This took a while and was accomplished by the publication of Ganopolski and Brovkin (2017) and Willeit et al. (2019). Now, some of the earlier findings are also corroborated with the new CLIMBER-X model.

*Furthermore, as I will discuss below, there are still some uncomfortable theoretical hurdles. For example,*
*the linear relationship the derivative of ice volume (in m3/s) and the orbital forcing (in W/m2) is not straightforward to explain from the theory of ice sheets.*

The Referee here challenges my Model 2, which is described in Appendix A4, and contrasts it with the theoretically derived model described in Verbitsky et al. (2018) (hereafter V18), and not in my favour. I had the privilege to discuss the V18 paper on several occasions with the first author, Michail Verbitsky,
and I know this work well. However, I do not believe that the governing equations of this model are derived from theory and not just postulated. Being heavily involved in the modelling of ice sheets and climate-ice sheet interaction over the past 20 years, I am not aware of the existence of any "theory of ice sheets" which can be used to derive a single equation describing the mass balance of all ice sheets on the planet (or even just a single one). The complexity of the problem is demonstrated in Fig. 1, which
depicts the mean annual surface mass balance of the northern ice sheets at 25 ka and 15 ka. A short glance at this figure is sufficient to recognize that there is no simple relationship between surface melting and the area of ice sheets. In the real world, the melting occurs only over a narrow ablation zone, which usually occupies a small fraction of the total area of ice sheets, but the ablation area rapidly expands during critically important parts of glacial cycles – glacial terminations, as shown in Fig. 1 below.

[Figure]

**Fig. 1**. Surface mass balance in cm/yr at 25 ka (left) and 15 ka (right) from the ONE_1.0 experiment with the CLIMBER-2 model (Ganopolski and Brovkin, 2017).

My approach is fundamentally different from V18. When developing Model 2, I did not try to derive model equations from the "first" principles. Instead, I choose the simplest possible equation which allowed me to reproduce main CLIMBER-2 results, first of all, its stability diagram in the phase space of orbital forcing, which is characterized by multiple equilibria and bifurcation transitions (Fig. 1 in Calov and Ganopolski, 2005). Note that a very similar stability diagram was reported by Abe-Ouchi et al. (2013). Assuming that the ice area is proportional to ice volume (the correctness of this assumption is described below), a simple equation for ice volume evolution can be written in the form (Appendix A4):

$$\frac{dv}{dt} = av - bv^2 - cf \ ,$$

where $v$ is ice volume and $f$ is orbital forcing. Again, unlike V18, I extensively tested this equation against the CLIMBER-2 result (see Fig. 2 below), and the results were quite satisfactory. In particular, a linear relationship between ablation and orbital forcing works well, which is not surprising because the correlation between total surface mass balance and orbital forcing is 0.8. Admittedly, the linear relationship between the derivative of ice volume and the orbital forcing is not my invention - it was used, for example, in P98 model.

*In summary, I would concede that it is adequate to use the word "theory" in the title but I would be more explicit about the focus on Quaternary glacial cycles and on the fact that this contribution is part of a long-term development.*

These issues I already addressed above.

Here is my proposal:

*A proposal: "steps towards generalizing the Milankovitch theory of Quaternary ice ages"*

Below I explain why I cannot agree with this proposal.

*I leave it with the editor, based on the comments of the other reviewers.*

Concerning the Referee's suggestion to change the title to "*steps towards generalizing the Milankovitch theory of Quaternary ice ages*". Unfortunately, the suggested title does not correctly represent the essence and significance of my paper.

First, about "steps toward generalizing". In my manuscript, whether the Referee likes this or not, I present not some steps towards generalizing, but the Generalized Milankovitch Theory. Obviously, the Referee does not place me in the cohort of Titans which are allowed (without any "steps" and "toward") to use such ambitious terminology as "Generalized theory", "Unified theory" or "the global theory" (these are the subtitle of the book and, the titles of Section 3 and Chapter 14 correspondingly of Saltzman's "Dynamical Paleoclimatology"). Still, I believe that the choice of the title in this special case (medallist paper), the choice of the title of the manuscript is the prerogative of the author.

Concerning the Referee's suggestion to specify the name of the theory, namely, "Milankovitch theory of Quaternary ice ages". Firstly, such a term does not exist. Google Scholar found zero matches for this term, and Google found only two, but, ironically, the first match is my GMT paper, which does not contain such a term.

Secondly, the term "Quaternary ice ages" is anachronism. While "ice ages" were considered relatively short climate events during the Milankovitch times, we now know that the Quaternary is essentially a continuous "ice age" with short interruptions.

Thirdly, GMT is the general theory of glacial cycles, not only Quaternary glacial cycles. Glacial cycles began before the Quaternary (2.55 Ma), but what is even more important, GMT is also applicable to the future -both "natural" and Anthropogenic (Archer and Ganopolski, 2005; Ganopolski et al., 2016; Talento and Ganopolski, 2020). Thus, GMT is not a "Quaternary theory".

And last but not least. Do we really need a clarification for the "Milankovitch theory"? Such respected authors like A. Berger, J. Imbrie and D. Paillard used the term "Milankovitch theory" without any additional clarifications. Such a clarification would only be needed if this term is ambiguous, in other words, if there is not one but several different Milankovitch theories. Of course, Milankovitch's contribution to science is not restricted to glacial cycles, and André Berger in his recent paper even named Milankovitch "the father of paleoclimate modelling". It is also true that Milankovitch had much more serious ambitions than just solving the "ice age problem". The the title of Part V of the Canon (in English translation) contains "The mathematical climate of the Earth". However, to everybody who really read Canon, it is obvious that the knowledge about climate processes during the Milankovitch era was absolutely insufficient for the development of the theory of climate. The real theory of climate began to be developed only with the advent of powerful computers and other modern technologies in the 1960s. At the same time, Milankovitch indeed made an important contribution to the understanding of glacial cycles. Not only did he clearly formulate the problem and choose the right forcing, but he also developed a simple model which accounted for the key climate feedbacks and used this model to test the hypothesis about the astronomical origin of glacial cycles. He even made an attempt to validate his modeling results against geological data available at that time. Thus, the name "Milankovitch theory" is well deserved, but there is only one Milankovitch theory – the theory of glacial cycles.

 **Line by line comments**

*p. 2: 50: "The comprehensive theory cannot be simple". It is a nice punchy sentence but perhaps a bit tautological.*

Yes, this was an ironic statement addressed to colleagues who believe that glacial cycles can be
explained by one or two ordinary differential equations. The author would be happy to remove this sentence.

*In fact, can we ever have 'a' general comprehensive theory of ice ages? Perhaps this would be similar to ask for a general theory of the blood circulation. One can convincingly sketch the general idea (this is one objective of "Model 3"), and then go into endless details about the dynamics of fluids, chemistry, function*
*and shape of heart etc. There will never be "one" theory because we speak about a complex system, which can be attacked from many angles.*

Concerning (im)possibility to develop a theory of complex systems. Admittedly, I know nothing about blood circulation. However, I do know that in other scientific disciplines, there are many generally accepted theories, such as general relativity theory in physics, valence bond theory in chemistry, theory
of evolution in biology, plate tectonic theory in geology, etc. What is so special about climate science, and why does the Referee think that it is impossible to develop a theory of glacial cycles which will be comprehensive and (eventually) generally accepted?

The correct theory, even if it is not yet comprehensive, is very useful (and the wrong theories will die out on their own). We already do have one nearly generally accepted theory of glacial cycles – the
Milankovitch theory. Just imagine that this theory would not exist, and we would not have a clue of what drives glacial cycles. Where would we be now in understanding past climate changes if the present-day orbital parameters of Earth's orbit would be fixed in program codes as was the case in the first generations of climate models?

Now, a few words about complexity. The author is well aware of the complexity of the Earth system as
he devoted most of his 40-years scientific career to modelling the Earth system in general, and glacial cycles in particular. However, complexity is not an obstacle but rather the stimulus for developing a comprehensive theory. The task of the theory is to provide a general framework (concept) useful for understanding the observed (past) phenomena and predicting new (future) ones. To deal with the full complexity of the Earth system, we have Earth system models (EMICs and complex GCM-based ESMs).
Significant progress in modelling and understanding glacial cycles has already been achieved with EMICs (i.e. Ganopolski and Brovkin, 2017; Willet et al. 2019). Now, the turn is for complex ESMs.

Where I really see a problem related to the complexity of the Earth system is in deriving simple models from 3D equations describing the dynamics of climate, ice sheets and their interactions.

*My suggestion (first draft): "Despite significant progress in understanding climate dynamics and [...]*
*glacial cycles, questions have remained about the mechanism of importance of glacial dynamics, carbon cycle, and scaling relationships between different variables,*

I am not sure what "*mechanism of importance*" means; possibly this is a typo, but the importance of ice sheet dynamics and climate-carbon cycle feedbacks for glacial cycles is generally acknowledged, and a few remaining issues which are not yet fully understood are extensively discussed in my manuscript. As far as "scaling relationships" are concerned, this is not a part of GMT, and I suspect that this issue is of interest only to a small cohort of mathematical climatologists.

*and how they concurred to synchronise glacial cycles on short eccentricity cycles.*

The GMT gives the answer to this question.

*These are the targets for progressing towards better theories of glacial cycles.*

Unfortunately, I do not understand what "better" means in this context and why "theories" are in plural form? Does the Referee believe that we need several alternative theories of glacial cycles? I think that one correct theory would be enough.

Action: I will add a sentence addressing the remaining question formulated by the Referee.

*Somewhere in the text I would also see adequate to pay some tribute to the visionary paper by MacAyael*
*1979.*

The MacAyael (1979) paper will be cited.

*l. 125, typo (starting with ".")*

Will be fixed

*Section 3.2 : Perhaps this is a place to note a difference of status between different forms of low-order*
*models. Paillard 1999 is clearly "inductive" in the sense that the existence of three states, plus the associated time scales, emanate from inspection of the data. Paillard 1999 summarises what he sees in the form of a small model, and then draws the consequences of it. Verbitsky et al. 2018 start from theoretical considerations about glacial scaling laws and deduces the ice volume trajectory, with obviously a bit of (reasonable) fine tuning. Saltzman's work from the late eighties /early nineties are*
*somewhere in between, depending on how one looks at it. These different models have different functions, all useful,  in the construction of our understanding of the astronomical control of ice ages.*

(Again, it should be Paillard 1998). I agree that it makes sense to make a distinction between purely inductive and partially inductive conceptual models of glacial cycles. This will be done in the revised version of the manuscript.

*l. 296: the word "massive" may be unnecessary.*

Agreed, "*massive*" will be removed.

*ll. 320 - 322 : CLIMBER-2 itself was tuned; it implicitly includes observations, in that sense it is not quite certain that CLIMBER-derived constraints should be considered as fully independent.*

I agree with the Referee that "independent" is not the right word in this context, and it will be removed.
What is meant here is that Model 3 formally contains five "tunable" parameters, which is a bit too much for such a type of model aimed at reproducing a single wiggly curve. However, three of these parameters ($f_1$, $t_1$ and $t_2$) can be directly derived from a combination of paleodata and CLIMBER-2 results with reasonable accuracy. Thus, only two tunable parameters remain.

*l. 404: "erasing the memory". this is correct, but the phenomenon is already implicitly there in MacAyael*
*1979.*

The MacAyeal paper will be cited. Again, I didn't claim that GMT is based solely on my own ideas.

*l. 449: "Model 3 represents" -> "Model 3 is"*

The sentence will be modified

*l. 481: The author may also consider a reference to the Huybers - Tziperman 2008 paper.*

As will be discussed below, this paper is based on erroneous approach. This is why it will not be cited. It is enough that original Huybers 2006 paper is cited.

*l. 605: I must admit having been unconvinced about the "Quantum tunnel" analogy (in which case the potential barrier is crossed by a form of delocalisation). For the modelling of ice ages, the basic limitation of the potential barrier image is that it is 1-dimensional, while the dynamics for the relaxation imply at*
*least another degree of freedom. This is not what happens in quantum dynamics. I leave the author with these considerations without any intention to fight on this point.*

Of course, any analogy, especially analogy with quantum mechanics, is limited. I use this analogy to stress the fact that glacial termination ("catastrophe") in my models is not related to the crossing of bifurcation point (B2 in Fig. 4). Presently, there is a widespread misunderstanding (both in mass culture
and science) that crossing of a bifurcation point ("tipping point") should lead to an imminent catastrophe. By using an analogy with the tunnel effect, I want to demonstrate that glacial termination is a completely different type of instability than that associated for example with Stommel's model of thermohaline circulation or Budyko's temperature hysteresis.

*ll. 650 - 655 : Basal sliding related to thermal balance (as encoded by Verbitsky et al. 2018) is another*
*potentially important mechanism not mentioned here. From informal conversations with glaciologists, I understand that this is a very plausible, major ingredient for the deglaciation catastrophe.*

The Referee is mistaken: the role of basal sliding is not only discussed (L. 631) as part of the domino effect but it is also depicted in Fig. 13a by a big blue arrow ("*fast sliding*"). And, of course, basal sliding only occurs "*when the temperature at the base of ice sheets reaches the pressure melting point*" (L. 802).
However, basal sliding is not a specific feature of deglaciations. As shown in Ganopolski et al. (2010) (Fig. 6c), during most of the last glacial cycle, 20-30% of the ice sheet base was at the pressure melting point and thus was sliding. However, it makes a big difference whether sliding occurs over a hard bed (rocks) or soft bed (sediments), because "*the thick terrestrial sediments make ice sheets more mobile since temperate ice (ice at the pressure melting point) moves much faster over sediments than over bare rocks*"
(L. 743). This is why, only when the northern ice sheets prior to LGM spread over large areas covered by unconsolidated sediments, basal sliding becomes an important factor affecting ice sheet stability. Of course, there are several other mechanisms which likely also contributed to the "deglaciation catastrophe", such as marine ice sheet instability and "proglacial lakes ice sheet instability", which are also discussed in sub-section 5.5 Glacial terminations: the domino effect.

*p. 24 before section 5.8: The author may consider adequate to mention already at this point the Paillard-Bouttes theory about sequestration and release of carbon due to change in the formation of AABW (it is alluded to later in the text, admittedly).*

The importance of enhanced AABW formation due to brine rejection and its impact on carbon sequestration in the glacial ocean was discussed already in Brovkin et al . (2007), i.e. two years before
the first Boutess et al. paper. This is the citation from Brovkin et al. (2007): "*Enhanced formation and*

*increased density of AABW water masses is primarily caused by more extensive sea ice formation with associated brine rejection around Antarctica*" (PA4202). The difference between our modelling results and "Paillard-Bouttes theory" is that, in our version of the CLIMBER-2 model, brines are effectively mixed with the surrounding water masses, while Bouttes and co-authors used the same model but proposed that most of the brines arrive into the deep ocean without mixing. As an expert in ocean modelling, I was always sceptical about the realism of such an assumption. Nonetheless, in Ganopolski and Brovkin (2017) we tested the mechanism proposed by Bouttes and co-authors and concluded that the assumption about the penetration of a large fraction (>50%) of brines into the deep glacial ocean leads to results incompatible with proxy data.

*p. 26 ll. 820 - 824 : regolith over CO2. Fair point, but pre-800 ka CO2 estimates remain uncertain, especially that one needs a trend over the mean state (set by the balance between outgassing and weathering), which may differ from what individual glacials or interglacials estimate.*

I fully agree with the Referee that pre-ice core reconstructions of CO2 concentration are not as reliable as ice core data, but I cannot see how this fact contradicts to what is written in the manuscript: "*there is* **no strong evidence** *for the decline of CO2 level directly prior to the MPT (Hönisch et al., 2009; Chalk et al., 2017; Yamamoto et al., 2022)*".

*l. 866 : The long interglacials indeed bring an interesting constrain in the decision between 'self-sustained' vs 'driven' constraints and this section is relevant. However, what model simulations give in this respect (l. 867) may bring a tautological argument, especially when it comes for future climate simulations because the anthropogenic CO2 emissions would have broken the self-sustained oscillation, if there was one, anyway.*

I agree that the last sentence of the paragraph can be interpreted in a way that a future extra-long interglacial disproves the mechanism of self-sustained oscillations. Of course, we do not have data from the future. The idea was to say that the mechanisms of forced and self-sustained oscillations have very different implications for future glacial cycles. This sentence will be modified accordingly.

*sect. 5.11 : The discussion in this paragraph is reasonable, but I would like to use this opportunity to clarify one point. Several models with self-sustained oscillations have indeed sensitive dependence on parameters and/or additive stochastic noise, in the sense that certain terminations may be triggered one precession cycle in advance, or delayed, delaying the whole sequence. Mathematically, this indeed occurs as a manifestation of 'non-chaotic strange attractors' (Mitsui and Aihara, 2014; Crucifix, 2013). But in all cases, these models do not display sensitive dependence to _initial conditions_. I agree that the surprising efficiency of 'simple rules' suggest that the timing of deglaciations is less sensitive to details or stochastic elements than these models may suggest, but on the other hand, I observe that the catastrophic character of some deglaciations (especially termination V) is hard to capture by those models which are the most well-behaved (like CLIMBER). So we have to explain a paradox here: On the one hand terminations would be highly catastrophic (which indeed suggest a domino effect, in essence very sensitive to an initial trigger), and on the other hand their timing would be very robustly set by the astronomical forcing.*

The Reviewer mixed here predictability with model performance, which are different things. For example, many GCMs still simulate spurious double ITCZ, but this does not mean that the correct seasonal evolution of ITCZ is, in principle, "unpredictable". Similarly, the fact that in one of four runs, the "well-behaved" CLIMBER-2 with interactive $CO_2$ (Willeit et al., 2019) failed to simulate complete deglaciation during Termination V, does not imply that Termination V is "unpredictable". It only means that a weak orbital forcing at the time of Termination V is too weak for a given model to trigger deglaciation in 100% of cases. CLIMBER-2 is the first earth system model which simulates glacial cycles with interactive $CO_2$ and ice sheets. I am sure future models will be even better, and the problem of Termination 5 will be gone. It is also noteworthy that even in the single CLIMBER-2 run, where Termination V wasn't completed, the next glacial termination (TIV) was successfully simulated. Thus, a single model failure does not prevent successful simulations of the next glacial cycles, and this is despite the fact that CLIMBER-2 simulate strong internal (random) variability associated with millennial-scale AMOC transitions and Heinrich-type ice surging events (Ganopolski et al., 2010). Thus, CLIMBER-2 results strongly suggest that glacial cycles are mostly deterministic and thus predictable. This is related to the fact that, according to GMT, glacial cycles of the last 1Ma are tightly locked to the 100-kyr eccentricity cycle. This also explains why a simple rule (subsection 5.13) is so successful. Obviously, the situation would be completely different if the durations of individual glacial cycles were a random alteration of 2 or 3 obliquity cycles, as has been proposed by other workers.

*l. 960: it is quite clear what the author means here, but perhaps the semantics could be polished ( "simply by a non-linear response" , "an arbitrary non-linear response", "a very special type of non-linearity"). Perhaps there is a way to nail it a bit more explicitly. The key point is that a linear response (or weakly non-linear) response to a non-linear transformation of the orbital forcing will generate all the eccentricity spectrum, including its 100ka, 400ka and even longer components, simply because it will merely rectify the signal. What we need is a strongly non-linear internal dynamics, that is, internal feedbacks (or, to put it otherwise: dynamical mechanisms) which are triggered by the internal state of the system. Perhaps one could contrast 'non-linear dynamics' to 'non-linear response' but I concede that this is not quite satisfactory either.*

The appearance of all eccentricity frequencies in the system response to orbital forcing is not related to whether the response is weakly or strongly non-linear. For example, Model 3 is strongly non-linear: it has multiple equilibria and bifurcation transitions. However, without the termination regime (k=2 in Eq. 2), Model 3 produces results very similar to the weakly non-linear Imbrie and Imbrie model, and the spectrum of simulated ice volume contains all eccentricity frequencies very much alike Fig. A3. Thus, modelling the 100 kyr world requires not just a strong nonlinearity but a very special type of nonlinearity. In Model 3, this very special type of nonlinearity is related to the Termination regime and explained by i) the existence of quasiperiodic forced oscillations with periodicities close to the shortest period of eccentricity, ii) an independence of the magnitude of these oscillations on the amplitude of orbital forcing; and iii) a weak sensitivity of the timing of glacial terminations on the amplitude of orbital forcing. As far as a discussion of internal feedback and other "dynamical mechanisms" is concerned, it is not applicable to Model 3, which only describes the mathematics of glacial cycles. The physics of glacial cycles is described in the next section, Section 5, where I discuss in detail the physical processes and feedbacks related to the entire glacial cycles and specifically to the termination regime (the domino effect).

*There is one fundamental difficulty with many conceptual Quaternary ice sheet models, and, admittedly,*
*many low-order ice sheet models (include from this reviewer, see Martínez Montero et al. 2022): why*
*would the ice volume derivative be proportional to its volume?*

I guess it shouldn't. At least in my models, unlike V18, I do not assume that ice volume derivation $dv/dt$ is proportional to ice volume (or ice area). In Model 3, the $dv/dt$ term (in the glaciation regime, k=1) depends only on the distance to the corresponding equilibrium branch, i.e. $v-V_e$, not $v$. In Model 2, which
was used to derive the phase portrait for Model 3, $dv/dt$ is related to the ice volume but in a strongly nonlinear manner since the equation for ice volume it contains a quadratic term, while in the deglaciation regime, the relationship between $dv/dt$ and $v$ is even more complex.

*Indeed, if we admit that mass balance is area times net accumulation rate, it would perhaps be natural to*
*except the net accumulation rate to be proportional to climate factors, which themselves are excepted to*
*be proportional to the _area_ (not volume) of the ice sheets.*

The assumption that the total mass balance of the ice sheet (accumulation – ablation – calving) is proportional to the area of the ice sheet, as it is postulated in V18, does not have a solid scientific basis. The total ablation is indeed approximately proportionality to the ice sheet area, and calving is reasonably well correlated with the ice volume (Fig. 2), but the key for simulation of glacial cycles is ablation. As
discussed above (see Fig. 1) there is no reason to expect that total ablation is directly related to the total ice sheet area since it strongly depends on the geographical position and the width of the ablation zone.

[Figure]

Fig. 2. Relationships between (top left) ice volume vs ice sheet area, (top right) ice area vs global accumulation in
Sv=$10^6$ m$^3$/s, (bottom left) global calving vs ice volume, (bottom right) anomaly of orbital forcing vs surface mass balance derived from Ganopolski and Brovkin (2017) ONE_1.0 experiment. Greenland ice sheet is excluded in all cases.

*These considerations concur to the scale relationship of Verbitsky et al. 2018 (dS/dt proportional to*
*S^(3/4)\*accumulation, S the area). Yet, I agree, the linear relationship between bulk accumulation and*
*volume works well in simple models, and this is perhaps why, besides its simplicity, it is so popular. The*
*problem is briefly alluded to in Verbitsky and Crucifix, CPast 2023 (original version in*
*https://doi.org/10.5194/cp-2023-30). At least there is a little theoretical challenge, here.*

The Referee assumes that, since the scaling relationship between volume and ice area is derived
theoretically, it must be correct. In fact, this scaling relationship is only applicable to the cold-based,
equilibrium ice sheets. Real ice sheets are not completely cold-based and are not in equilibrium. As a
result, the simplest relationship $A \sim V$ works better than the "theoretical" $A \sim V^{3/4}$ most of the time (Fig.
2a), and during some (usually short) periods, neither of the two relationships really works. One example
are the glacial inceptions, when ice area increases much faster than the ice volume (Fig. 5 in Willeit et al.,
2023).

**Interglacial metrics**

*The difference between eq. A1 and the original Huybers metric (l. 1034) is n\*I_0, where n is the number of*
*days with I>I0. We indeed expect this latter term to be a function of obliquity, which then cancels part of*
*the obliquity component in the original Huybers metric. Both the Huybers original formulation, and the*
*Milankovitch caloric insolation, work in similar ways.*

This statement is not correct. Caloric insolation, as any other metric for "summer insolation" obtained by
integration over a certain period of time (it does not matter whether this is one day, one month or six
months), contains a strong precessional component, while Huybers' metric does not, because precession
is nearly completely eliminated from this metric by varying period of integration. (Hot summers are short
summers). Fig A1 in my manuscript demonstrates a clear difference between spectra of caloric half-year
insolation (Fig. A1h) and Huybers' summer insolation (Fig. A1i). It is also noteworthy that the corrected
version of "integrated summer insolation", which, in terms of Huybers and Tziperman 2008 paper,
should be written as $\sum_{d=1}^{365} \beta_d(\Phi_d - \tau)$, contains even more precession than the caloric half-year summer
insolation (Fig. A1j).

*Huybers and Tziperman 2008 are quite explicit about why they do not clip the insolation (just after their*
*equation 2);*

Indeed, Huybers and Tziperman 2008 tried to justify Huybers' original idea and wrote: "*Note that the full*
*magnitude of the insolation intensity, not only the portion above the threshold, is summed under the*
*assumption that most incident radiative energy will eventually lead to ablation once the freezing point is*
*obtained*". With that, they demonstrate a lack of understanding of the processes controlling the surface
mass balance of ice sheets. According to such a theory, surface melt must suddenly jump from zero to a
large value the next day after insolation crosses threshold value τ. Of course, this does not happens in
reality. In reality, at the time when insolation crosses this threshold, the absorbed insolation is
completely compensated by the net outgoing longwave radiation plus (usually less important) sensible
and latent heat fluxes. This is why only the insolation above τ (i.e. $\Phi_d - \tau$), not $\Phi_d$, can contribute to snow/ice melt during the entire melt season. Thus, the day after crossing the threshold, the amount of insolation available for melting of snow/ice will be not 300 W/m$^2$ but just a few W/m$^2$!

*I would like to bring the additional arguments:*

*- when we calibrate one of the good old Saltzman's models we obtained posterior distributions with about equal weights of precession and obliquity (Carson et al., 2013, Figs. 3 and 5, compare gamma_P and gamma_E)*

*- in LOVECLIM, with interglacial conditions, precession and obliquity have more or same-order-of-magnitude effects on the "GDD" (but actually equivalent to PDD) at high latitudes (Bouncer et al., 2015,*
*Fig. 7) with, I would concede, more precession in North America.*

Are these arguments in defence of "integrated summer insolation"? But they obviously do not serve this purpose. The Referee wrote that orbital forcing contains comparable contributions of precession and obliquity, and I agree with that. As Fig. A1 in my paper shows, all metrics for summer insolation, including the corrected version of "integrated summer insolation" contain both strong obliquity and
precessional components. Only Huybers' version of "integrated summer insolation" contains practically no precession. This is precisely the reason why Huybers published this metric, since in the early 2000s, he denied the important role of precession in glacial cycles and speculated that "*the ice sheets terminated every second or third obliquity cycle at times of high obliquity*" (Huybers and Wunsch, 2005).

*So the discussion clearly has merit and the author is undoubtedly right in tackling this issue, but perhaps*
*the conclusion lines 1046 deserve some caveat.*

Although I put the discussion of "integrated summer insolation" in the Appendix, I believe this is an important issue for two reasons:

1) If "integrated summer insolation" has a physical meaning, then it would represent an elegant solution for the "41-kyr world problem", namely, the lack of precessional signal in paleorecords prior to MPT.
However, because "integrated summer insolation" is a wrong metric for summer insolation, and all "right" metrics contain strong precession, the 41-kyr problem remains. In particular, in Willeit et al. (2019) simulated ice volume prior to MPT, although is dominated by obliquity, but still contains precessional component.

2) The motivation for Huybers' 2006 paper is clear: he was looking for a metric which does not contain
precession and found one. Unfortunately, he made a mistake. Everybody can make a mistake. However, the fact that some workers still use this wrong metric is worrisome.

This is why I see no reason to "soften" the conclusion of Appendix 1.

***The Domino effect***

*Such a domino effect is indeed what is encoded, more or less explicitly, in many conceptual models, but how 'irreversible' or 'catastrophic' the domino needs to be is not straightforward. Some deglaciations are deeper than others, or sometimes stalled (those leading to 7e, or the strange 15c /15 a duet) which conceptual models tend to overdo.*

The complete deglaciation (i.e. final $v$=0) for simplicity is encoded only in Model 3 but not in Model 2. In
particular, Model 2 correctly simulates "incomplete" deglaciations (without any parameter changes)

during several pre-MBE interglacials (Fig. A5). Actually, with complex models, it is easier to get incomplete rather than complete deglaciations. In any case, "completeness" or "incompleteness" of deglaciations (whatever it means) has nothing to do with the concept of the domino effect: some ice always remained on the Earth during the Quaternary.

Brovkin, V., Ganopolski, A., Archer, D., and Rahmstorf, S.: Lowering of glacial atmospheric CO2 in response to changes in oceanic circulation and marine biogeochemistry, Paleoceanography, 22, PA4202, 2007.

Willeit, M., Calov, R., Talento, S., Greve, R., Bernales, J., Klemann, V., Bagge, M. and Ganopolski, A.: Glacial inception through rapid ice area increase driven by albedo and vegetation feedbacks. EGUsphere, 1-41, 2023.

---

## Author Comment (AC2)

I thank the Referee for the useful and constructive comments and suggestions. Below is my response with the original Referee's comments in blue italic.

*1 . **I do not agree with all the views** presented here by the author.*

5    I suppose this means that the Referee disagrees only with some of my views. This interpretation is supported by the next sentence:

*I certainly do agree on the main mechanisms behind the 100-kyr-cyclicity.*

This Referee's statement is highly appreciated

*But I don't like the so-called "regolith hypothesis"*

10    This is a personal opinion of the Referee. However, I respectfully disagree with the following statement:

*now, even its first promoter (Peter Clark) explains why it was not a good idea after all.*

While the authors have the right to retract the paper (as far as I know, Clark and Pollard's 1998 paper was not retracted), they cannot retract their own hypothesis if it has been widely discussed for 25 years and has already gained strong support (e.g. Willeit et al., 2019).

15    Now, about Peter Clark and the fate of his hypothesis. As far as I know, Clark presented his new opinion first at EGU 2021 ("Requiem for the Regolith Hypothesis"). I did not attend his online presentation at EGU, but I attended in person the QUIUGS workshop in Lamont in September 2022 where Clark presented "Requiem" again. My personal impression was that Clark's presentation wasn't met very enthusiastically, and a number of critical questions have been asked. Now, three years after the
20    "Requiem" abstract was submitted to the EGU meeting, the paper has still not been published, and it is impossible to learn based on which methodological advances Clark disproved his former hypothesis. This is why Clark's personal opinion on this issue is of no relevance for my paper.

*In the paper, it is clearly presented as a hypothesis, which fills our lack of knowledge on the origin of long-term trends: may be, for completeness, the author could state that this remains a controversial*
25    *hypothesis.*

I believe any hypothesis in some sense is controversial, i.e. it is not generally accepted. After the hypothesis is finally proven and universally accepted – it is no longer "hypothesis"; it is a Law or a Theory. Of course, there are a number of other ideas about the nature of MPT, and in the last paragraph on page 26 I cited five papers presenting alternative views (Chalk et al., 2017; Farmer et al., 2019;
30    Hasenfratz et al., 2019; Ford and Raymo, 2020; Sutter et al., 2019). Since my paper is not a review paper, I think five references are enough. In addition to my paper, there is a 20-page review by Berends et al. (2021), which describes many more ideas about the nature of MPT.

*2 . Figure 14 is not referenced in the text. Besides, in my opinion, it is entirely useless and does not help the reader to understand the paper. I would suggest removing it.*

35    Agreed. The figure will be removed.

*3 . When discussing the speed of the forcing behind the MPT, ie. the rate of change of the critical ice volume Vc, around lines 815-820: an interesting paper was written on this specific point by Legrain,*

*Parrenin and Capron (Nature communications Earth & Environment, 4, 2023) using a rather similar "threshold based" model; with the conclusion that a gradual change appears more likely when using random parameters.*

40

I will cite Legrain et al. in section 4.6 in the sentence (L. 428) "*To reproduce the MPT in P98, the critical ice volume was made time-dependent with a smaller value at the beginning of Quaternary and a larger one toward the present.*"

*4. Equation (2) line 300. A minus sign is missing for k=2 (v should decrease during terminations).*

45     Thank you! Indeed, this is a very unfortunate typo, which will be fixed.

*5. line 574. I am not convinced at all that the 100-kyr-cycle is a "peculiar regime". Such a 100-kyr periodicity appears in many different contexts and in many pre-Quaternary Earth's paleoclimatic records. For instance, in Pälike et al. (Science 2006) there is a clear 100-kyr cycle in the $^{18}O$ that might be linked to Antarctic ice-sheet variations. Of course, interpretations are more difficult for these earlier periods, but*
50     *the Quaternary is certainly much too short a time span to talk about "peculiar" or "ordinary" regimes.*

The presence of eccentricity periodicities in the paleoclimate records prior to the Quaternary is not surprising but rather expected: any nonlinear transformation of orbital forcing (and Earth is such a nonlinear transformer) should cause the appearance of ALL eccentricity frequencies. This is precisely what is seen in Pälike et al. (2006) and some other records, and which is very similar to the frequency
55     spectrum of the Imbrie and Imbrie model but not in the real spectra of the late Quaternary (Fig A3). What is unique about the Late Quaternary is the absolute dominance of one eccentricity frequency (100 kyr) and the absence of others. Such a situation requires a very "peculiar" type of non-linearity, and such behaviour is not seen prior to the Late Quaternary. Of course, "peculiar" does not mean that this never happened during the entire Earth's history, but I have never seen anything similar in the earlier
60     paleocrecords.

*6 . line 560: « phase locking  has a different meaning … ». Of course, "model MiM" and "model3" are different, but in both cases the locking is directly linked to threshold crossing. It is not clear to me why « phase locking… » should have a different meaning.*

Fully agree, "*different meaning*" is not the right expression – phase locking is a phase locking. What I
65     meant to say here is that the **mechanisms** of phase looking in MiM model (as well as in different Van der Pol oscillators, etc.) and in Model 3 (also Model 2, Paillard 1998) are different: in the first case, the self-sustained oscillations with a periodicity close to 100 kyr exist without any external forcing and the amplitude modulated extremal forcing under certain conditions can synchronise these internal oscillations with eccentricity cycles. In the second case, the models do not have any internal oscillations,
70     and glacial cycles only arise under the influence of precession-obliquity forcing. Under some conditions, these cycles are synchronised with the amplitude modulation of the precessional cycle. I will modify the manuscript to make this point clear.

*7.  Figure 8e: there is a single very long "cycle" of about 150-kyr in the histogram. I am wondering which one it is… and how is it possible with this model: a few details on this particular cycle could be helpful.*

75     The reviewer is absolutely right – a single long cycle depicted in the histogram (Fig. 8e) is not present in Fig.8b. This is because the experiments shown in Fig. 8 and 9 began at 3 Ma from initial $v=0$ conditions, but only the last 2.8 Ma were shown and analysed due to a strong dependence of the model solution from

the initial conditions during the first 0.2 Ma. However, because of an algorithmic mistake, when producing a histogram shown in Fig. 8, the entire 3 Ma run was analysed, and at the beginning of this run (prior to 2.8 Ma), there is one 150-kyr long cycle. For consistency with the rest of Fig. 8 and 9, this single long cycle will be excluded from the histogram, and Fig. 8e will now look as shown below:

[Figure]

In addition, to be precise in the description of this experiment, the sentence (L. 400/401) will be modified: "To enhance the resolution of frequency spectrum, the model has been run through the past 3 Ma, of which the last 2.8 were analysed and shown in Fig. 8 and Fig. 9".

*8. Line 350: "Climber-2 has a problem simulating timing of TV while Model 3 does not"… I find this quite interesting! In the author's opinion, is it pure chance? Or could conceptual models be "in some way" more robust than physical models?*

Yes, this is an interesting point which deserves some elaboration. Indeed, during MIS12 to MIS11 transition, Model 3 overperforms CLIMBER-2. This is because the orbital forcing during Termination V is very weak and under such forcing CLIMBER-2 (with interactive CO2) simulates TV too late or even fails to simulate complete deglaciation (Willeit et al., 2019, Fig. 2). Model 3 does not posses such problem because the conditions for triggering glacial termination in the models are: $v > v_c$, $F > 0$, $dF/dt > 0$, where $F$ is the anomaly of orbital forcing. This explains why Model 3 is insensitive to the amplitude of orbital forcing. However, if I change the last condition to $dF/dt > 5$ W/(m² kyr), Model 3 results become similar to CLIMBER-2, as it skips Termination V. Since this did not happen in reality, the minimum value of $dF/dt$ required for triggering glacial termination must be rather small. Interestingly, simulations of other glacial terminations of the Late Quaternary are much more robust.

*9. some typos:*

*Figure 12: Qusi-linear ->Quasi-linear*

*Legend Fig 7: artifitial -> artificial*

Thank you! These typos will be fixed